# Short-term particulate matter contamination severely compromises insect antennal olfactory perception

Qike Wang[1,2,5], Genting Liu[1,2,5], Liping Yan[1], Wentian Xu[1], Douglas J. Hilton[2], Xianhui Liu ®[3], Wenya Pei[1], Xinyu Li[1], Jinbiao Wu[1], Haifeng Zhao ®[4], Dong Zhang ®[1] ✉ & Mark A. Elgar ®[2]

The consequences of sub-lethal levels of ambient air pollution are underestimated for insects, for example, the accumulation of particulate matter on sensory receptors located on their antennae may have detrimental effects to their function. Here we show that the density of particulate matter on the antennae of houseflies (*Musca domestica*) collected from an urban environment increases with the severity of air pollution. A combination of behavioural assays, electroantennograms and transcriptomic analysis provide consistent evidence that a brief exposure to particulate matter pollution compromises olfactory perception of reproductive and food odours in both male and female houseflies. Since particulate matter can be transported thousands of kilometres from its origin, these effects may represent an additional factor responsible for global declines in insect numbers, even in pristine and remote areas.

The detrimental impacts of anthropocentric pollutants on organism health, fitness, and population viability have been extensively documented for wildlife – from plants to vertebrates[1-4]. Particulate matter (PM) might be even more dangerous than other common air pollutants such as $NO_x$ or ozone, yet its ecotoxicological effects on many types of organisms including insects[5], and on ecosystems more generally remain relatively unclear[6]. Insects accumulate PM on the body surface and this might cause toxic effects to them[7-10]. PM includes a mixture of solid particles or liquid droplets suspended in the air and is produced from both natural and anthropogenic sources[11]. PM is one of the predominant air pollutants in urban environments[12,13], but is nevertheless recorded in high concentrations beyond these sources: over 40% of the global landmass is exposed to annual PM concentration that exceeds World Health Organization recommendation of annual average concentration ($<10\,\mu g/m^3$ (ref. 14), Fig. 1a). Surprisingly, these areas include many remote, comparatively pristine habitats and ecological hotspots (Fig. S1a). PM is highly heterogeneous, and differs in sources, morphology, elemental composition, and particle size[15]. There is evidence that $PM_{10}$

$(2.5\,\mu m < particle\ size \leq 10\,\mu m)$ has more inorganic or metal components, including toxic heavy metal elements, and $PM_{2.5}$ contains more organic pollutants such as benzene and polycyclic aromatic hydrocarbons[5,16].

The effects of particles on insect reproduction function were first documented in the early 20th Century[17]. Since then, several studies have shown that ingestion of food contaminated by PM could have detrimental impacts on the development, reproduction, and longevity of different insects[10,18-21]. Recent correlational studies document a potential impact of chronic exposure to extreme air pollution levels over a long period of time on insect physiological and developmental functions[8]. However, unambiguous links between insect fitness and the frequency and level of ambient air pollution are still lacking. For example, as a representative of the major cities around the world that suffers from air pollution, the PM levels in Beijing fluctuate dramatically between March and October, when adult insects are most active (Fig. S1b), but the duration of air pollution episodes (AQI, air quality indicators ≥100) typically last less than a few days and are much shorter than the life span of insects such as honey bees and houseflies

¹School of Ecology and Nature Conservation, Beijing Forestry University, 100083 Beijing, China. ²School of BioSciences, University of Melbourne, Melbourne, VIC 3010, Australia. ³Department of Entomology and Nematology, University of California Davis, Davis, CA 95616, USA. ⁴Faculty of Architecture, University of Melbourne, Melbourne, VIC 3010, Australia. ⁵These authors contributed equally: Qike Wang, Genting Liu. ✉ e-mail: ernest8445@163.com

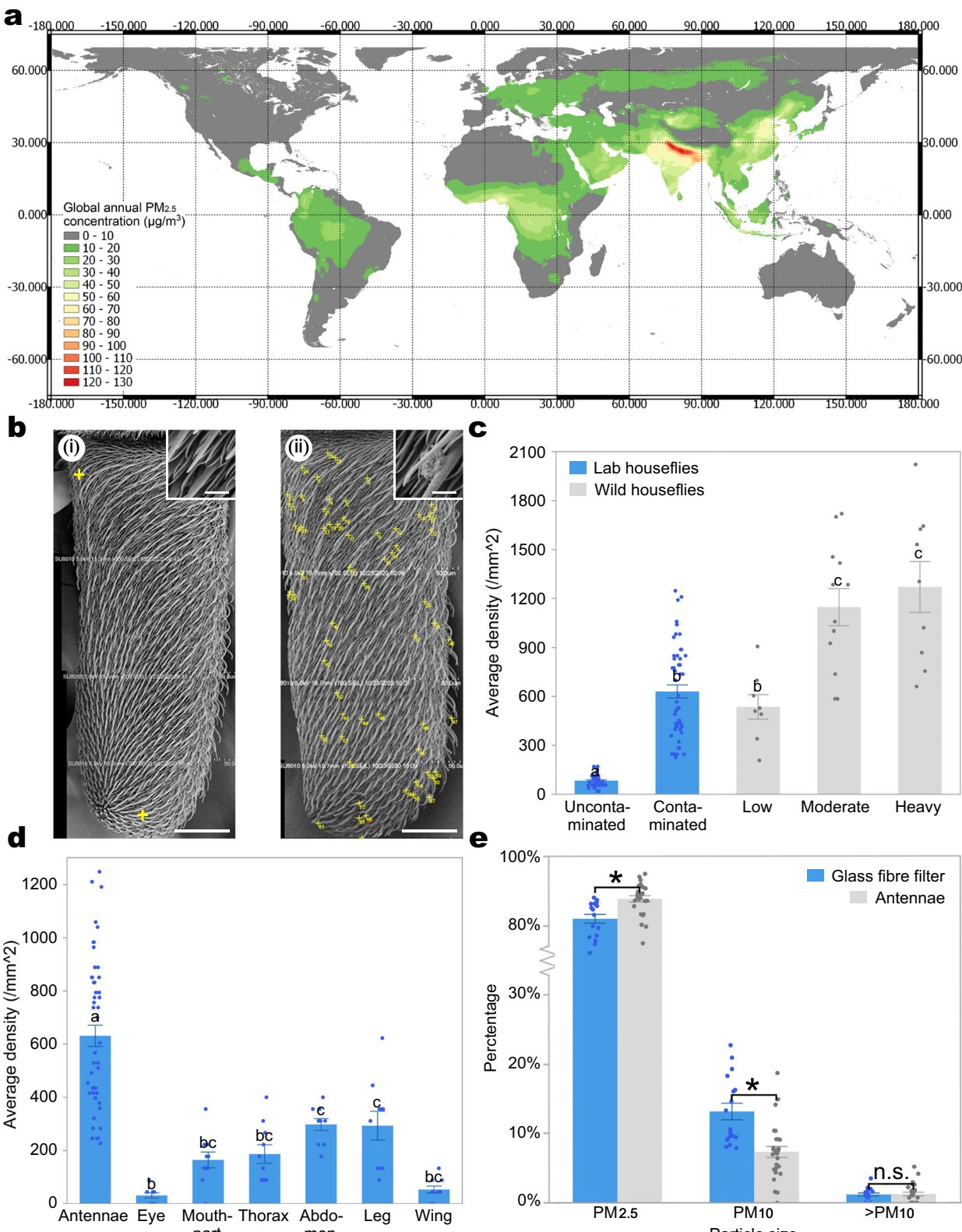

(Fig. S1c)[22]. Currently, there is little, if any, information about the effects of the more frequent short-term and lower pollution level exposure events, which happens more frequently and commonly under natural conditions.

Odour molecules, used by insects to locate food resources, oviposition sites and reproductive partners, are perceived only when they physically interact with olfactory receptors, typically but not exclusively located on antennal sensilla[23,24]. Accordingly, optimising the capacity for odour perception acts as a strong selection pressure shaping the evolution of insect antennal morphology[25]. Even subtle differences in micro-morphology can influence the movement of odours across the antennae[26]. Similarly, the number and density of sensilla on insect antennae are thought to reflect those selection pressures[23], and a reduction in sensilla number may compromise signal

**Fig. 1 | Particulate matter (PM) contaminated houseflies. a** Global annual average $PM_{2.5}$ concentration on different landmass (except for Antarctica) from 2015 to 2019 – ~40% of the landmass is exposed to an annual PM concentration that exceeds WHO recommendation of annual average <10 μg/m³ (Data source: https://sites.wustl.edu/acag/datasets/surface-pm2-5/#V4.GL.03). **b** Stitched SEM micrographs showing that (i) uncontaminated housefly antenna has much less PM than (ii) contaminated antenna, with yellow crosses marking the distribution of individual particles. **c** Average density of PM detected on the antennal surface of uncontaminated, experimentally contaminated, and wild houseflies captured in low (AQI ≤ 50), moderate (50 < AQI ≤ 100), and high (100 < AQI ≤ 150) pollution levels. Experimentally contaminated housefly antennae have a significantly higher density of PM than uncontaminated antennae, and the density is comparable to that collected in an urban environment in Beijing (Table S3, generalised linear mixed models with Tukey post hoc test, $F_{4,123} = 43.25$, $P < 0.001$, $n = 25$). **d** Average density of PM detected on different body parts of contaminated houseflies: the antennae has significantly higher PM density than any other body parts (Table S4, generalised linear mixed models with Tukey post hoc test, $F_{6,114} = 18.41$, $P < 0.001$, $n = 10$). **e** Higher percentage of $PM_{2.5}$ was found on the antennal surface than on fibre glass filters (Table S5, Wilcoxon Test, $PM_{2.5}$: $P < 0.001$; $PM_{10}$: $P < 0.001$; >$PM_{10}$: $P = 0.835$, Antennae: $n = 27$; Fibre glass filter: $n = 17$). Scale bars: $b = 50$ μm, 5 μm in box. Different lower letters and asterisk indicate significant difference between groups, centre: mean, error bars: SE. All $p$-values are based on two-sided tests. Source data are provided as a Source Data file.

perception[27,28]. While the placement of sensilla on antennae increases their efficiency in olfactory reception, it may also leave them vulnerable to airborne contaminations like PM. For example, the scales on moth antennae can reduce the possibility of contamination on sensilla by diverting the airflow away from the antennae[26], but this may not be possible for insects, such as bees and flies, with antennae directly exposed to the airflow.

Here, we use a combination of scanning electron microscopy (SEM), energy dispersive X-ray spectroscopy (EDX), behavioural assays, electroantennography (EAG) and transcriptomic analysis, to demonstrate that PM accumulation on the antennae of houseflies (*Musca domestica* L.) compromises their capacity to detect fitness-related odours.

## Results

### PM contaminates housefly antennae

We initially collected houseflies from the urban area of Beijing (Haidian District), when air quality indicators (AQI) were low (AQI ≤ 50), moderate (50 < AQI ≤ 100), and high (100 < AQI ≤ 150). Our SEM images revealed an accumulation of PM on their antennae, which increased with the levels of ambient air pollution (Fig. 1b, c). Subsequent experimental exposure, which controlled for insect age and the duration of exposure, revealed that the density of particles on the antennae of lab-reared houseflies exposed to 12 h ambient air pollution (100 < AQI ≤ 150) was around seven times greater than that of houseflies maintained in clean air, and comparable with wild specimens collected from low to moderate levels of pollution (Fig. 1c). Following controlled exposure to air pollution, PM accumulated across the head, thorax, legs, and abdomen of houseflies (Fig. S2) but was significantly more densely distributed on the antennae (Fig. 1d). Combined SEM and EDX analysis of housefly antennae and fibre glass filters confirmed four common types of PM: silicate, sulfurate, fly ash, and metal particles (Supplementary Note 1 and Fig. S3). Interestingly, a higher percentage of $PM_{2.5}$ was found on the antennal surface than the ambient PM captured by a Particulate Matter Collector on fibre glass filters, which provided a relatively unbiased sampling of the PM in the air (Fig. 1e). This suggests that $PM_{2.5}$, which is generally considered more hazardous for human health[5], may pose a greater threat to the health of houseflies than larger PM components.

### PM compromises the ability of houseflies to detect odours

We explored whether the accumulation of PM on the antennae of male and female houseflies compromised their ability to detect and respond to crucial chemical signals and cues: conspecific female odour and food odour (Fig. 2). Y-maze olfactometer assays revealed that PM contamination on the antennae consistently compromised the capacity of male and female houseflies to respond to food odour cues, and of male houseflies to respond to female conspecific odours (Fig. 2b). In all three experiments, a non-random proportion of the uncontaminated houseflies entered the arm of the maze leading to the odour, across the entire range of dilution levels, indicating that they were consistently attracted to the odour. In contrast, a statistically significantly smaller proportion of contaminated houseflies responded to the odour, and this proportion indicated that these houseflies were selecting the arms of the Y-maze at random (Fig. 2b-d).

### PM damages the sensitivity of housefly antennae

We then used EAG assays to confirm that the differences in the response of contaminated and uncontaminated flies in the Y-maze assays reflected a diminished capacity to detect the odours. The response to the key sex pheromone component (Z)-9-Tricosene, measured as the adjusted voltage change, was significantly more prominent in uncontaminated than contaminated antennae of both female and male houseflies (Fig. 3a, b). Similarly, the adjusted voltage change of the uncontaminated antennae in response to the food attractant stimulus was significantly lower in the contaminated groups of both female and male houseflies (Fig. 3c, d). This effect persisted over 10 days after the exposure treatment for females but not for males (Fig. 3e, f), so a relatively brief exposure to air pollution at a young age can nevertheless have long-term effects on key behaviours of females, including mating (3-4 days old) and searching for food and oviposition sites (7-8 days old)[29].

### PM changes gene expression level in houseflies

To compare the impacts of PM on housefly gene expression, we performed transcriptomic analysis on the contaminated and uncontaminated houseflies in spring and summer, when houseflies are most active in Beijing. PM impacts gene expression pattern in antennae more severely for female than male houseflies in the spring, with 395 differently expressed genes (DEGs) in the antennae for females and 80 for males (Fig. 4a). In the summer, PM impacted the antennae of males more severely than females: 898 DEGs in the females and 1313 DEGs in the males (Fig. 4a). Significantly more DEGs were observed in the summer for both male and female houseflies than in the spring (Fig. 4a). Forty-nine overlapping DEGs were between male and female antennae in the spring, while 348 were overlapping in the summer (Fig. 4b). Interestingly, only four DEGs overlapped among the four groups, suggesting that the impacts of PM on houseflies differed between the sexes and seasons. All DEGs in male and female antennae were subjected to Kyoto Encyclopedia of Genes and Genomes (KEGG) and gene ontology (GO) enrichment analysis to identify the most modified pathways in housefly antennae (Fig. 4c, d). This reveals that PM pollution has a potential impact on metabolic pathways including glucose (glycolysis and citrate cycle), amino acid, nucleotide, fatty acid metabolism, and circadian rhythm.

Olfactory perception related pathways are under-represented in the KEGG database for houseflies, so we also performed a focused analysis with respect to genes related to the important steps in the olfactory signal processing pathways including odorant binding proteins (OBPs), gustatory/olfactory receptors, ion channels and neuron signalling[30,31]. Surprisingly, only one differentially expressed gustatory/odorant receptor gene (*GR2*) was found in both male and female antennae, and the expression level of odorant binding proteins between males and females were generally different (Fig. 4e and

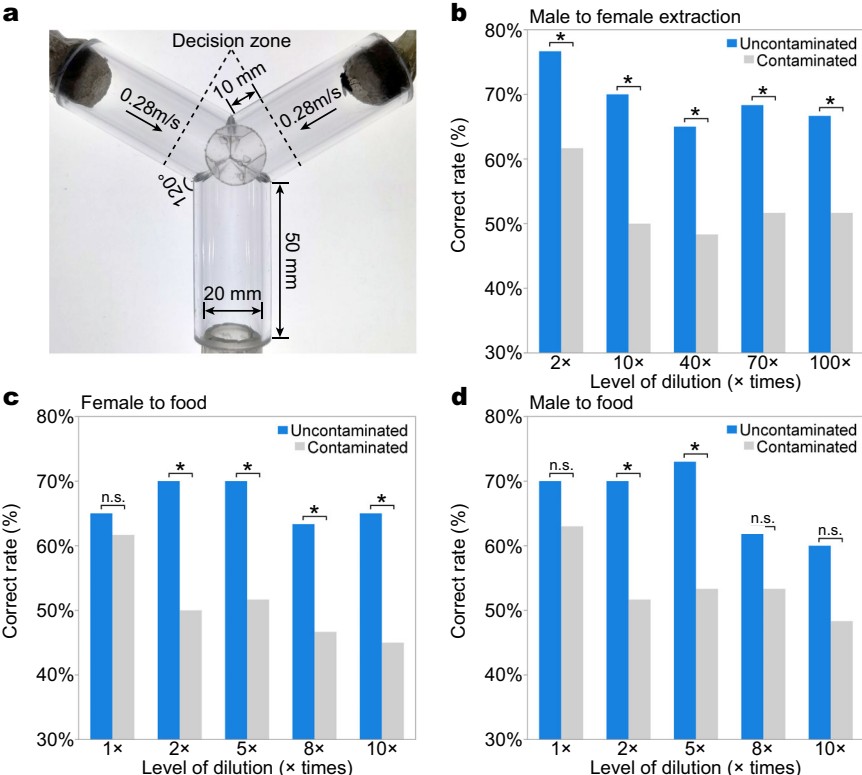

**Fig. 2 | Particulate matter pollution influenced olfactory behaviour of house-flies. a** Dimension and zoning of the olfactometer, arrows showing the direction of airflow. Houseflies were deemed to have made a choice after crossing the dotted lines, and we recorded the first choice only of the housefly in each assay. **b** The proportion of uncontaminated male houseflies that chose the odorous arm of the Y-maze containing female odour was significantly greater than that of con-taminated male houseflies, irrespective of the concentrations (Table S6, general-ised linear model: full model: $\chi^2 = 22.61$, df = 9, $p = 0.007$, $n = 60$). **c** The proportion of uncontaminated female houseflies that chose the odorous arm of the Y-maze containing food odour was significantly greater than that of contaminated female

houseflies, irrespective of the concentrations (Table S6, generalised linear model: full model: $\chi^2 = 20.44$, df = 9, $p = 0.015$, $n = 60$). **d** The proportion of uncontami-nated male houseflies that chose the odorous arm of the Y-maze containing food odour was significantly higher than that of contaminated male houseflies, irre-spective of the concentrations (Table S6, generalised linear model: full model: $\chi^2 = 17.38$, df = 9, $p = 0.043$, $n = 60$). $\chi^2$ tests were performed between each pair of contaminated and uncontaminated flies in **b**–**d**, and asterisk indicate significant difference between groups. All $p$-values are based on two-sided tests. Source data are provided as a Source Data file.

Supplementary Data 1–4). This indicated that perception of a range of odours could be jeopardised, further confirming the impact of PM on antennae that was highlighted by the behaviour assays and EAG[32,33]. Many of the DEGs belong to the same gene family, Cytochrome pro-teins (CYPs) P450 (Supplementary Data 1–4). These genes were gen-erally associated with cell detoxification, such as of pyrethroid or other synthetic insecticides[34]. The regulatory trend (i.e., upregulated, or downregulated) for most of these CYPs was different between male and female flies, again suggesting that the impact of PM differs between male and female antennae. Importantly, many of these CYPs were downregulated in either male or female fly antennae, suggesting the detoxification function in these flies could be jeopardised. Similar comparisons performed on the bodies of male and female houseflies revealed the possible impact of PM on several physiological processes in various tissues and organs (Supplementary Note 2, Fig. S4 and 5, and Supplementary Data 5 and 6).

## Discussion

Our experiments reveal the novel insight that only 12 h exposure to high levels of air pollution ($100 < AQI \leq 150$) significantly compromises the ability of male and female houseflies to detect fitness-related odours. This finding highlights a potentially severe impact of short-term, sub-lethal PM exposure to insects. Importantly, the effect was evident following a period of exposure that is much shorter than both the adult lifespan of houseflies and the duration of most pollution episodes in Beijing, so our data underestimate the true impact of PM

on insects in Beijing and surrounding regions. Insects further from sources of air pollution may also be affected by the accumulation of PM on their antennae, the rate of which will depend on multiple factors including the frequency, duration, concentration of air pollution epi-sodes, the chemical composition of PM, and the interaction between PM and different insect species.

Our EAG analysis confirms that the behavioural assays reflect deficiencies in olfactory perception, rather than poor health, but this does not discount other physiological effects of PM, such as through the digestive or respiratory systems[10]. Access to the respiratory system of houseflies is through the mesothoracic and metathoracic spiracles, and dense microtrichia around each spiracle could act as filters to prevent particles entering the trachea[35]. We detected very few particles around the perimeter of each spiracle (Fig. S2i, j), suggesting that PM are unlikely to enter through respiratory organs, as occurs in vertebrates[36,37].

PM may impact olfactory perception across different insect taxa and may be experienced in remote habitats far beyond major sources of air pollution (Fig. S1a, b). Indeed, our field surveys in the urban environment in Beijing and in rural habitats impacted by bushfire in Victoria, Australia revealed PM pollutants on the antennae of diverse insects, including bees, wasps, moths, and other species of flies (Fig. S6). The severity of impact across insect taxa may depend on both behaviour and antennal morphology: for example, species whose antennae are protected by scales may be less likely to be contaminated[26].

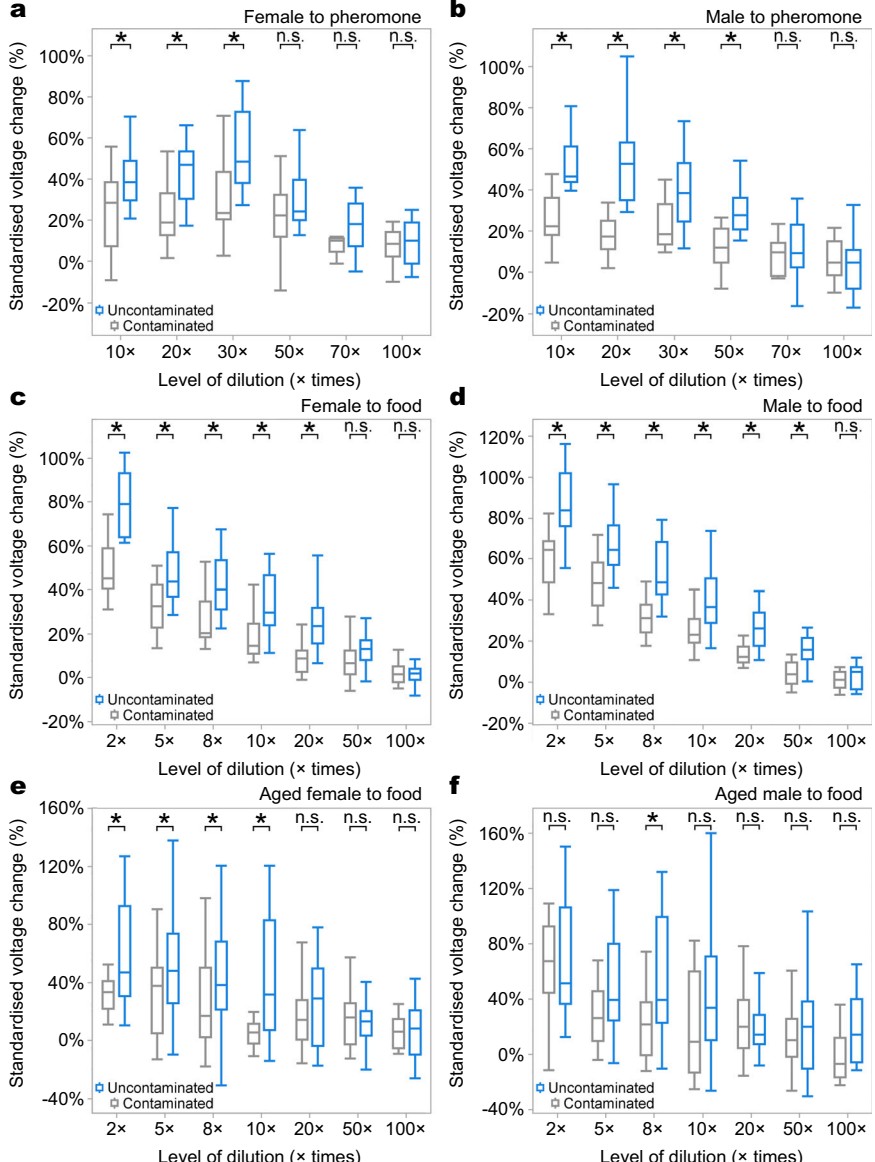

**Fig. 3 | Electroantennogram (EAG) assays showed that PM pollution influences antennal function of houseflies. a** Uncontaminated female and **b** uncontaminated male houseflies were more sensitive to most concentrations of the sex pheromone, (Z)-9-Tricosene, than contaminated counterparts (Table S7, generalised linear mixed models with Tukey post hoc test, Female: Treatment: $F_{1,169} = 11.16$, $P = 0.001$, $n = 15$. Table S7, Male: Treatment: $F_{1,169} = 36.48$, $P < 0.001$, $n = 15$); uncontaminated (**c**) female and (**d**) male houseflies were more sensitive to most concentrations of food odour than their contaminated counterparts (Table S7, generalised linear mixed models with Tukey post hoc test, Female: Treatment: $F_{1,197} = 63.19$, $P < 0.001$, $n = 15$. Table S3.4, Male: Treatment: $F_{1,197} = 47.90$, $P < 0.001$, $n = 15$); **e** aged (7–10 days after PM exposure) contaminated female houseflies were not as

sensitive to three of the four different concentrations of food lure than aged uncontaminated counterparts (Table S3.4, generalised linear mixed models with Tukey post hoc test, Treatment: $F_{1,256} = 19.26$, $P < 0.001$, $n = 20$); **f** the response of aged males was overall independent of contamination (Table S7, generalised linear mixed models with Tukey post hoc test, Treatment: $F_{1,194} = 0.26$, $P = 0.622$, $n = 20$). The lower and upper hinges of the box plots correspond to the 25th and 75th percentiles, the central line represents the median, and the whiskers indicates the ranges; "*" indicates significant difference and "n.s." non-significant differences between treatment groups. All $p$-values are based on two-sided tests. Source data are provided as a Source Data file.

Grooming behaviour (self-cleaning) including the antennae surface is widespread in insects[38], but in flies, it may not protect their antennae against PM contamination. Flies use the thick bristles on their fore tibia for removing dust or debris on their heads and antennae, and with a diameter of 8–12 μm, the bristles are much thicker than PM$_{2.5}$ (Fig. S7). Although our experimental flies could groom themselves *ad libitum* both during and after the PM exposure, a significantly higher number of PM were observed on the antennae of the contaminated flies (Fig. 1d). The frequency of grooming behaviour was similar between the contaminated and uncontaminated flies, and the level of contamination on the fore tibia was similar between these groups

(Fig. S7). These data suggest that grooming behaviour may be less effective against PM particles, especially PM$_{2.5}$.

Insects play crucial roles in providing fundamental ecosystem services through the regulation of pests and diseases, pollination, and nutrient cycling[39], all of which require effective and efficient detection of chemical cues. The diversity and abundance of terrestrial insects are declining across almost all biogeographic regions[40], suggesting that the function and stability of both natural and anthropogenic ecosystems are at risk[41–47]. The reasons for this massive disappearance of insects are thought to derive from a complex combination of habitat changes and anthropogenic pollutants[48], perhaps also

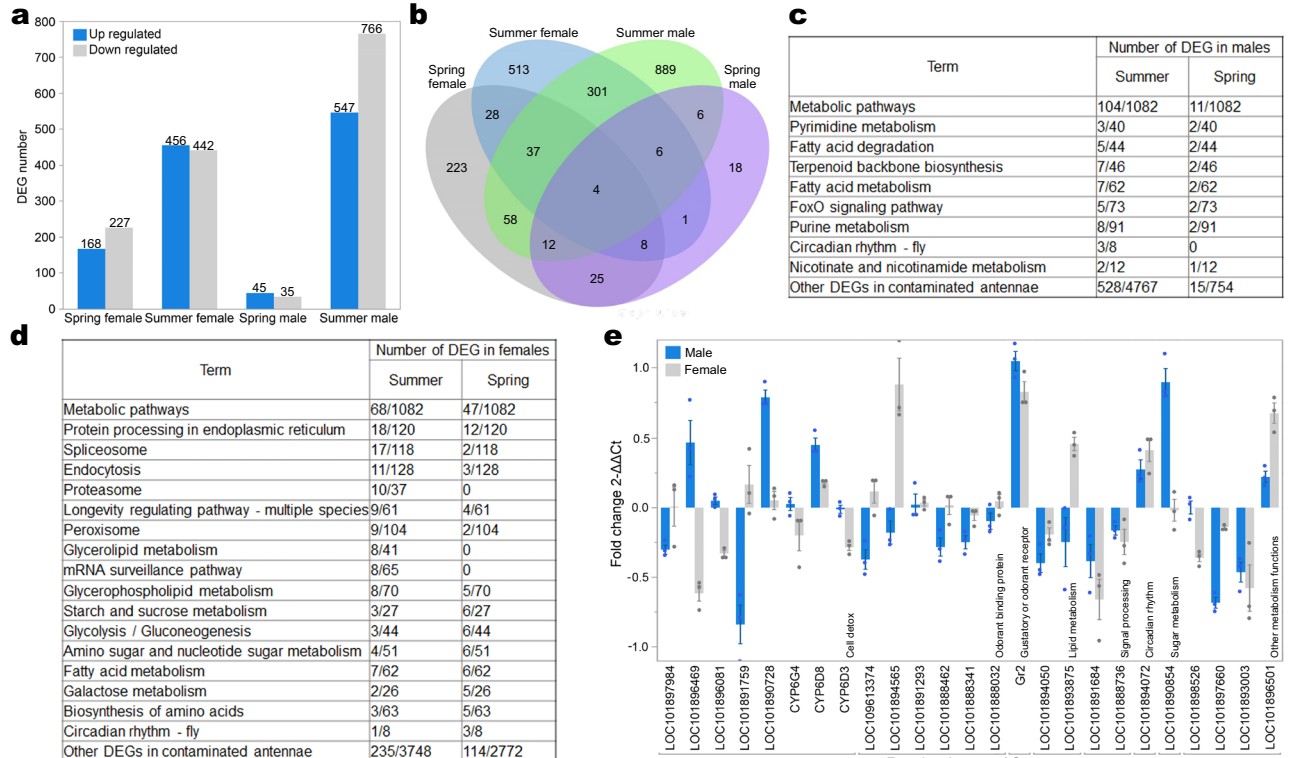

**Fig. 4 | Transcriptomic analysis showed that PM influence gene expression level in the antennae of houseflies. a** The number of differently expressed genes (DEGs) identified in the antennae of male and females experimentally contaminated in the Spring and Summer compared to uncontaminated flies (DEGs are defined as Bonferroni adjusted $P \leqslant 0.05$, p-values are based on two-sided tests). **b** A Venn diagram of the number of DEGs between female and male houseflies experimentally contaminated in the Spring and Summer (see Supplementary Data 1–4). KEGG pathway enrichment results of (**c**) males and (**d**) females showed the pathways with the highest number of DEGs (number of DEGs/number of genes in each term; see Supplementary Data 1–4). **e** Expression levels of genes important in the function of male and female antennae were validated by RT-qPCR. PM contamination affected the antennae of male and female flies differently in most of the validated genes (n = 3). Centre: mean, error bars: SE. Source data are provided as a Source Data file.

including the effects of air pollution on odour detection. More systematic sampling and assays in a wide range of habitats are needed to assess this possibility. The consistency of our data makes a compelling case to investigate the potentially widespread impact of air pollution on insect pheromone and cue perception, which will affect mating and foraging success. For example, even sub-lethal impacts of air pollution may diminish population viability of insects if males fail to detect females, leading to female mating failure and a reduction in population density, an effect that may be especially significant in smaller populations[49,50].

## Methods

### Documentation of PM on insect antennae

Wild houseflies (*Musca domestica* L.) were captured between June and July 2020, when Air Quality Index (AQI) conditions in Beijing were defined as low (AQI ≤ 50), moderate (50 <AQI ≤ 100), and high (100 <AQI ≤ 150), according to US-EPA 2016 standards[51]. The ambient AQI measures in Beijing were obtained from The World Air Quality Project[52] (Fig. S1b, c), and the indoor and outdoor AQI were confirmed by a Hanvon PM detector M1 (Hanwang Technology Co., Ltd, Beijing, China). The houseflies were carefully pinned and air-dried individually in a lab equipped with air purifier (Samsung AX70J7000WT Samsung Co., Ltd, Gyeonggi-do, South Korea) to prevent cross contamination before preparation for SEM (Scanning Electron Microscope) images.

We sampled PM from the polluted air using a glass fibre filter fixed in a Th-150c medium flow Atmospheric Particulate Matter Sampler, paired with a PM-100 multi-stage Particulate Matter Collector (Wuhan Tianhong Environmental Protection Industry Co., Ltd, Wuhan, China). Samples were obtained from the campus of Beijing Forestry University

for 2 h during the period of insect exposure (see below) with a flow of 100 L/min. Clean, glass fibre filters were prepared as controls.

Housefly pupae were sourced from the laboratory population maintained at the National Institute for Communicable Disease Control and Prevention, Chinese Centre for Disease Control and Prevention. The insects were cultured at $25 \pm 1\,°C$ with relative humidity of 30–50%, and 10:14 h light: dark photoperiod; the larvae were reared on a diet of water, wheat bran and milk powder mixture at the mass ratio of 200:100:1; and adults fed with aqueous 10% sucrose solution *ad libitum*. Pupae and adult flies were housed in a laboratory fitted with an air purifier (Samsung AX70J7000WT Samsung Co., Ltd, Gyeonggi-do, South Korea) that maintains both the $PM_{2.5}$ and $PM_{10}$ concentration below 5 µg/m³ (AQI < 20). We removed the deodorisation component of the air filter, thereby ensuring, as far as possible, that gaseous pollutants (other than PM) in the laboratory environment were similar to that outside the building. Male and female adults were separated shortly after eclosion, randomly allocated to treatments, and maintained in mesh cages (35 cm × 35 cm × 30 cm) until the experimental trials.

We controlled for variation in the length of exposure to air pollution by creating two experimental groups of flies: those that had been exposed to ambient air pollution (contaminated) and those that had not (uncontaminated control). Houseflies, <2 days post eclosion, were sexed and randomly allocated to identical flight cages, each containing about 150 individuals, which were then randomly assigned to the contaminated and control treatments. All the experimental behavioural assays were conducted between March and November when houseflies are active in Beijing. Contaminated flies were exposed for 12 h to the outside ambient air in treatment laboratories with open

windows in various locations in Beijing (Table S1), on days with high AQI levels (101-150), ambient temperatures at 15–25 °C, and between 7 am to 7 pm when the flies are most active. The exposure time was determined as our pilot study indicating that 12 h exposure yielded PM densities that were broadly similar to those obtained from wild caught houseflies. Control houseflies were transferred to a separate laboratory located next to the treatment laboratory with the similar size, orientation, temperature, and light conditions as the treatment laboratory, but fitted with an air filter that maintains an AQI < 20. The allocation of laboratory spaces to treatment or control was assigned randomly for each collection yielding a total of 35 batches (pairs of cages, see Table S1).

The houseflies were supplied with sufficient water and food (except for the food odour behavioural assays, see below). The flight cages were made from clean cardboard boxes of 80 cm by 40 cm by 60 cm, with five sides replaced by mosquito nets that allows sufficient aeration and activation room for the flies. To avoid any possible cross contamination, each flight cage was used once only. This procedure was conducted for batches (pairs of cages) of newly emerged flies of each generation (See Table S1), and the flies were sampled within 24 h for subsequent tests (unless specified otherwise).

We collected 25 houseflies from each of the laboratory control and contaminated groups (see above), and wild flies during light, moderate and high pollution level to compare the density of PM on the antennal surface using SEM ($n = 25$ each). Wild flies were collected from the campus of Beijing Forestry University at high pollution levels at the same time as the contaminated laboratory flies. The wild flies collected during the light and moderate pollution levels were collected at the same place but different pollution episodes during the same season. The two antennae of each individual were carefully mounted on the SEM stubs with one the anterior side up, and the other with the posterior side up. We additionally obtained the head, thorax, abdomen, a front leg, and a wing of each of five individuals from the contaminated laboratory group. Finally, we prepared a sample (5 mm × 5 mm) from the centre of the clean and the contaminated glass fibre filters (see above). All samples were mounted on aluminium stubs with double-sided adhesive tape, placed in a glass desiccator for 48 hours to dry thoroughly. The specimens were coated with gold in a spatter coater for 45 s (estimated gold thickness of 5 nm) and imaged on a Hitachi SU8010 field-emission SEM (Hitachi Corp., Tokyo, Japan) with the acceleration voltage of 5 kV, captured using a secondary electron detector (SE), and the scan speed of 40 S per frame.

SEM images of each antenna were stitched in image J (Fiji) using the "Tile" function from four smaller overlapping images (see Fig. 1b). The density of PM on each antenna were calculated by counting visible PM on the antennal surface, defined by the image, and divided by the area of each antenna. We counted the PM on both sides of the antennae of each individual. The density of PM on the glass fibre filters were counted by taking ten 150 μm by 150 μm quadrats (22500 μm²) on each filter. We used a total of 120 samples to calculate the density of PM ($n = 15$). All the PM were then classified by size (>$PM_{10}$, $PM_{10}$, and $PM_{2.5}$)[6] and the density standardised to (numbers)/mm². PM < 0.1 μm were too small and impractical to separate from the antennal sensilla surface features and thus were not included in this study. The average densities of PM on other body parts (eye, mouthpart, thorax, abdomen, leg, wing) of the contaminated and control groups of houseflies were calculated by counting all PM within four 150 μm by 150 μm quadrats taken from each of three to four individuals (each representing 22500 μm²) haphazardly placed within each image.

To conduct the point elemental analyses on the PM accumulated on the antennae and glass fibre filters, we used the Energy Dispersive X-Ray Spectroscopy (EDX) paired with a Hitachi SU8010 SEM. A series of common elements identified in PM in previous research (C, N, Na, Mg, Al, Si, P, S, Cl, K, Ca, Ti, Cr, Fe, Cu, Zn) were selected[22], and the voltage was set at 20 kV. Spot analyses were conducted on at least 100 particles randomly selected on the antennal surface of 50 individuals and glass fibre filters, and the blank area of each surface was measured to provide a baseline (Fig. S3).

## Behavioural assays

We used a Y-maze olfactometer to investigate the impact of PM contamination on the response of houseflies to food and conspecific female odours (Fig. S8). The Y-maze comprised three acrylic tubes (internal diameter 20 mm; length 50 mm) connected at 120°. The two arms were connected to clean odourless Teflon tubes (internal diameter 8 mm) that delivered either the odour stimulus or clean air control. The air was filtered with an activated carbon filter and an air scrubber, then pumped into the olfactometer at 0.28 m/s at the arm entrance through a long tube to maintain a laminar flow. After each trial, the Y-maze was cleaned with ethanol and detergent, and the position of odour stimulus and solution control were rotated to remove positional and arm effects for each trial. All trials were conducted at 20–25 °C, 40–60% humidity, with both arms of the Y-maze exposed to similar light intensity. Pilot experiments confirmed that the flies showed no preference when presented with the same odour at both arms.

We examined the response of houseflies to food (honey solution) and conspecific female odours (cuticle extraction of female flies, following Silhacek et al.[53]). Female cuticle extraction stock was prepared by soaking twenty 3–5 days old virgin female flies in 10 ml hexane for 1 min. We diluted the honey solution to 1, 2, 5, 8, and 10 times with water, and the sex attractant to 2, 10, 40, 70, and 100 times with n-Hexane. The odour treatment was delivered by applying 25 μL attractant solution to a piece of 10 mm × 10 mm filter paper, which was allowed to dry completely for at least 15 min. Twenty-five microlitre pure solvent (water for food odour assays, n-Hexane for female odours) applied onto a piece of filter paper was used as a control, which was also dried naturally for at least 15 min. Male and female houseflies were fed with water only before the food attractant assays, and only males were used for the sex attractant assays (females did not show any preference to this odour in preliminary analyses). Sixty males and/or females were tested on each dilution level of all experimental comparisons, yielding 600 trials for each assay. Each housefly was tested once only.

The test housefly was chilled at 4 °C for 60 s to reduce its activity level before being gently transferred to a cylindrical tube (35 mm long, 20 mm diameter, with mesh at one end) that was then inserted into the exit arm of the Y-maze. The trials were commenced when the recovered housefly entered the Y-maze olfactometer, and we recorded which arm of the Y-maze the housefly first entered, moving beyond 10 mm from the entrance to the arm (delimited by a line on the tube). Each trial lasted for 2 minutes with the observer being blind to the fly treatment, and we discarded trials in which the individual did not enter either of the arms of the Y-maze.

## Electroantennogram

Glass capillary tubes (outer diameter 2 mm) were pulled into micropipettes using a Dual-Stage Glass Micropipette Puller (PC-100, Narishige, Tokyo, Japan). The tips of the glass micropipette were polished to ensure the inner diameter of the glass electrode was slightly larger than the diameter of the housefly antenna. An Ag-AgCl electrode was placed in the glass micropipette that was filled with Ringer buffer (SL6438, Coolaber, Beijing, China) to record the antennal responses. The legs and wings of the chilled housefly were removed, and the individual was immobilised in a plastic micro pipette tip, with only the head exposed[54]. The two electrodes were positioned using micromanipulators under a stereoscope, with the indifferent electrode inserted into a compound eye and the recording electrode connected to an antenna, with the tip removed to form a complete circuit. EAGs were analysed and stored on the computer using EAGPro software

(Syntech, Hilversum, Netherlands). The procedure was deemed successful if a relatively stable baseline was observed on the recording equipment. The odour stimulations were delivered at the air speed of 0.5 m/s and duration of 0.5 s.

The odour was delivered through a steel nozzle (diameter: 6 mm; length: 15 cm) positioned 2 cm from the antenna (Fig. S9a). The purified and humidified air generated by a stimulus flow controller (CS-55, Syntech, Buchen-bach, Germany) was blown over the antenna at 12.5 mL/s. The stimulus flow controller generated air pulses through the odour cartridge at a flow rate of 10 mL/s, a compensating air flow was provided to maintain a constant current. We recorded the antennal response of 15 control and 15 contaminated houseflies (contaminated within two days of eclosion) of each sex to different concentrations of honey solution ($n = 15$) and 97% sex pheromone (Z)-9-Tricosene (Sigma-Aldrich)[55] ($n = 15$). Aged flies were maintained in the laboratory environment with the conditions described above for seven to ten days before EAG assays ($n = 20$). Similar rates of survival were observed between treatments, as over 90% of the flies in both groups survived. The honey was diluted with water to 2, 5, 8, 10, 20, 50, and 100 times (v/v), and the sex attractant was diluted with hexane to 10, 20, 30, 50, 70, and 100 times (v/v). Each housefly was tested against every dilution level of the stimulus. The baseline action potential and the response to solvent were also recorded for each housefly. Filter paper strips (5 mm×50 mm), adsorbed with 25 μL of the attractant solution and left until the solvent is dried completely for 3 min, was inserted into a glass Pasteur pipette to introduce the stimuli into the airstream[56]. Pasteur pipette and filter paper strips were used once only. Filter paper loaded with solvent was used as a control before and after presenting the stimuli, providing a baseline of each antenna. The stimuli were added to the air stream in increasing concentration at an interval of at least 30 s, to avoid sensory adaptation during recordings.

The recorded signals were amplified by an IDAC interface amplifier (IDAC-4, Syntech, Buchenbach, Germany), and the data analysed using Autospike 3.4 (Syntech, Buchenbach, Germany). We recorded the maximum antennal response of each trial, determined by the software EagPro (Fig. S9c). The responses were standardised as a %, using the formula: $\frac{100 \times (measured\ value - solvent\ response\ value)}{solvent\ response\ value}$.

## Transcriptome analysis and RT-qPCR

Adult male and female houseflies of the same age (48 h post eclosion) from both control and contaminated treatments were exposed to PM using the same protocol as the previous experiments in both spring and summer. Within 24 h after the treatments, the flies were removed from their cages and immediately frozen in liquid nitrogen for 5 min, and then stored at −80 °C. The antennae were excised (under a stereoscope) from the rest of the body using sterilized tweezers and placed in separate, sterilized cryopreservation tubes. Dry ice was used to ensure the temperature of equipment was maintained as low as possible. Each sample comprised 60 houseflies, and we obtained three biological replicates for each gender and treatment, yielding a total of 24 samples per season (three biological replicas × two body parts × two sexes × two treatments). The samples were stored at −80 °C until sequencing.

The total RNA was extracted using the TRIzol method[57]. RNA sequencing (RNAseq) was completed by Allwegene Technology Inc., Beijing. The cDNA library was then constructed using polymerase chain reaction (PCR) amplification. RNA-seq was performed with the PE150 sequencing strategy using an Illumina second-generation high-throughput sequencing platform. RNA-seq reads with inferior quality or adaptors were filtered. Clean read data were processed using Tophat2 and Cufflinks software to complete the alignment of transcriptomes. Differentially expressed genes and transcripts were then filtered for false discovery rate (FDR) adjusted P ⩽ 0.05. The reference whole genome library and annotation follows that of Scott et al.[58].

We assessed whether the impact of PM is associated with any changes in gene expression level by comparing genes that are associated with various physiological processes expressed in antennae and the rest of the body. The common up- and downregulated mRNAs were highlighted with volcano plots. We subsequently used GOseq[59] in gene ontology (GO) enrichment analysis to identify their functions, using the Kyoto Encyclopaedia of Genes and Genomes (KEGG). Volcano plots were generated using GraphPad Prism 8.0 (GraphPad Software, La Jolla California USA).

We selected 25 relatively highly expressed DEGs (differentially expressed genes) critical for the function of fly antennae, using RT-qPCR to validate the transcriptomic analysis results. Total RNA was extracted from three biological replica of housefly antenna and body samples, each containing 80 individuals, yielding 24 samples (three biological replica × two body parts × two sexes × two treatments). The first-strand cDNA was synthesised from 1 μg of total RNA extracted using PrimeScript™ RT reagent Kit with gDNA Eraser. RT-qPCR reaction contained 1 μl cDNA (100 ng), of 0.2 μl each primer (10 μmol), 5 μl 2x SYBR Green PCR buffer (Bio-Rad, Hercules, CA, USA) and 3.6 μl nuclease-free water. Primer pairs used for each gene are listed in Supplementary Table S2. The PCR conditions for the amplification action were as follows: 3 min at 95 °C, followed by 40 cycles of 10 s at 95 °C, 60 s at 54 °C, and 30 s at 60 °C. The PCR products were examined and calculated using the 2-ΔΔCT method. The relative expression was calculated against that of the house keeping gene GAPDH.

## Statistics and reproducibility

All statistical analyses were performed using JMP 14 pro (SAS, USA). For SEM data, we used generalised linear mixed models with Tukey post hoc test to investigate variation in the density of PM between different body parts of contaminated flies, and on the antennae between contaminated and wild flies, with individual identity included as a random effect (Table S3, S4). The proportion of each class of PM size on the antennae of the contaminated group and the glass fibre filters were compared using Wilcoxon test (Table S5). For behavioural assays, we used mixed models with a binomial distribution and logit link function to analyse the response of houseflies, with treatment (contaminated, not contaminated), dilution levels and their interaction as fixed effects and fly collection batch identity as a random effect (Table S6). For EAG data, we used mixed models with Tukey post hoc test to explain the variation in relation to blank value, with treatment (contaminated or uncontaminated) and dilution as fixed effects and fly collection batch and individual identity as a random effect (Table S7). We checked the distributions for normality of our variables before analysis. All statistical analyses employed two-tailed tests, with statistical significance set at <0.05, Bonferroni adjustment was made for multiple comparisons when it applies. Results from representative experiments are from at least 10 independently acquired individuals (Figs. S2, S3a-j, and S6).

## Reporting summary

Further information on research design is available in the Nature Portfolio Reporting Summary linked to this article.

## Data availability

The SEM, behaviour, EAG, and RT-QPCR data generated in this study have been deposited in the Open Science Framework at https://osf.io/v92xe/?view_only=0744a389866a4cdcb602ae967fe61960. The transcriptome data generated in this study have been deposited in the NCBI database under accession code PRJNA909937. The SEM, EDX, behaviour, and transcriptome data generated in this study are provided in the Supplementary Information/Source Data file. The world PM pollution level datasets are accessible via link https://sites.wustl.edu/acag/datasets/surface-pm2-5/#V4.GL.03, and the KEGG pathway database is accessible via link https://www.genome.jp/kegg-bin/show_organism?org=mde. Source data are provided with this paper.

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

## Acknowledgements

We thank Dr. Rui Wang and the Institute of Zoology, Chinese Academy of Sciences for the assistance with the electroantennograms; Ary Hoffmann and Devi Stuart-Fox for their helpful comments on earlier versions of the manuscript; Ben Phillips, John Morrongiello, Yunyun Gao, Yanning Wu, Jingquan Yan and Mingjian Zheng for statistical advice; Melanie Hammer for her help with the GIS data analysis; Carolyn De Graaf and Liam Salleh for their suggestions for interpreting the transcriptome analysis; Junna Shi for her assistance with SEM analysis; and National Institute for Communicable Disease Control and Prevention, Chinese Centre for Disease Control and Prevention for the source of houseflies. The National Natural Science Foundation of China (32170450, to D.Z.; 32201273, to Q.K.W.); Beijing Forestry University Outstanding Young Talent Cultivation Project (2019JQ0318, to D.Z.); Herman Slade Foundation (HSF1809, to M.A.E.); Australian Research Council (DP200101615, to M.A.E. and D.Z.).

## Author contributions

Q.K.W., G.T.L., D.Z. and M.A.E. conceived and designed the project; G.T.L., J.B.W., W.Y.P. and W.T.X. obtained SEM images and performed EDX analysis; G.T.L. performed behavioural and EAG experiments; G.T.L., Q.K.W. and M.A.E. analysed the behavioural and EAG data; G.T.L. collected transcriptome data; D.J.H, X.Y.L., X.H.L., L.P.Y. and Q.K.W. analysed and interpreted the transcriptome data; G.T.L., Q.K.W. and M.A.E. wrote the manuscript; H.F.Z. performed the GIS analysis; and all authors commented on the manuscript.

## Competing interests

The authors declare no competing interests.
