## [Peer Review File · Nature Communications]

Short-term particulate matter contamination severely compromises insect antennal olfactory perceptionReviewers' Comments:

Reviewer #1:

Remarks to the Author:

The authors of this study present a very interesting idea to investigate how air pollution can affect the olfactory system of insects. They combined different methods to investigate this effect from several angles. This data is novel and might have improved our understanding of air pollution effects on insect physiology and fitness. However, by using only two flight cages, the authors' experimental design cannot separate cage effects from exposure effects. The authors did not replicate the cages themselves, so the authors compared two non-independent groups of individuals to each other. They also do not mention the size of the flight cages and how many flies they put in each cage. Based on this, I think The authors cannot correctly analyse the experimental exposure data. However, the data from the wild-caught flies is still very interesting and could be written up for a more specific journal.

I also have a few minor comments:

Specific comments:

Line 173: please add how you obtained the AQIs here, i.e. move lines 208-211 up here.

Line 206: Did you only test this once in two cages? If yes, this means you only have one replicate for each treatment level, because flies are not independent of each other. Like this you cannot properly distinguish between the effect of cage and the experimental treatment of pollution exposure. Or did you repeat this several times?

Line 212: did you collect the wild flies at the same time as the exposed and control flies from your cages? Was this collection done in the vicinity of the cages? Please add this information to the method section.

Line 230: please add the software used for the statistical analysis.

Line 279-281: Does this procedure affect the EAG? Please add some reference showing that the removal of wings and feet does not affect the EAG read out. I am aware that the animal must be fixed somehow, and all flies were processed in the same manner. Nevertheless, it would be great to give the reader some idea of how invasive clipping of wings and feet is.

Line320: Please state here if you used 60 pairs of antennae per sample or 60 flies without antennae per sample. Or did you use both? Also, based on which parameter, the authors decided to only do three biological replicates with 180 flies/antennae pairs, rather than going for more replicates with fewer individuals per sample.

References 15, 28, and 38: Reference is in all caps.

Figure 1 a) please make sure you do not violate any copyright by using this map.

Please name the statistical tests performed to the f-statistics given in the figure legends.

Reviewer #2:

Remarks to the Author:

General comments

The manuscript entitled "Antennal contamination from short term air pollution severely compromises

insect olfactory perception" by Wang et al., represents an interesting and up-to-date effort to assess previously almost unstudied research niche. Indeed, despite the availability of the vast literature on the adverse effects of air pollution and more specifically particulate matter (PM) pollution on humans, there is less data about wildlife. The number of published research on insects is even less. This is the first reason, why the attempt of this study should be acknowledged.

Further on, the idea that a short-term air pollution episode might compromise the insects' olfactory perception and affect the antennae sounds reasonable. The PM pollution was indeed underestimated as a risk factor for insects' abundance and biodiversity. It wasn't also included in previous large-scale surveys. So, the manuscript suggests considering air pollution as a possible factor, contributing to the overall insect decline.

The multidisciplinary nature of the manuscript is also worth positive comments. The idea of using the combination of methods (SEM, EDS, EAG, qPCR, behavioral assays) is interesting and ambitious. This kind of study is needed and has the potential to improve our knowledge of insect responses to air pollution.

However, I have several concerns. First, there is a lack of understanding of the insect self-cleaning (grooming) behavior. It is only briefly mentioned in one place of the manuscript. However, it should be taken seriously during both experimental and writing stages in this kind of research. In my experience, driven from field behavioral tests, the first body part, which insect clean immediately, is ANTENNAE. They always clean those with mouthparts or/an anterior pair of legs. Did the authors conduct any kind of self-cleaning observations for their study species under natural and laboratory conditions? My second major notice is about literature coverage. In many places, the references are either too narrow (when describing a bigger picture) or not matching. The full text of the article should be familiar to the authors to give the proper reference, not only the abstract part (see my specific comments). Indeed, the literature on the topic is scarce, but it exists! I suggested some sources at the end of my report. The other concern is methodology. I've noticed several key issues in the SEM-EDX methods (use of the detector, PM density calculation). EDX spectrum is not presented in the supplementary data. It isn't clear why the density of PM was higher on the antennae than on the other body parts? I can't see any reasonable explanation for that. Also, the gene-expression part sounds too vague. The primer specification wasn't clearly explained. If the EAG was performed for alive insects with dissected legs and wings, all the data obtained are compromised by huge stress factors. This is fully true, that currently, there is very limited information on the effects of PM on insects' olfactory organs. But to confirm the real effects more detailed and structured methodology is needed. I will let the editor decide about the possibility to consider this manuscript for publication in "Nature Communications". However, all these key issues should be carefully addressed by the authors before the further step to publishing this data anywhere. This is an important study with the potential to generate high impact, but numerous issues should be corrected or redone first.

Specific comments

line 36-37: a bit odd formulation "density increases... with".

lines 43-44: Currently, the connection between 1st and 2nd sentences is weak. Probably, you can add one more sentence after the first one, briefly describing the types of anthropogenic pollutants that you mentioned at the beginning.

lines 44-45: "may also be affected" this formulation with probability sounds too vague, if you have the proof, avoid assumptions. I know the papers you have cited here and understand why you use this "may" mode. Indeed, the literature doesn't really confirm that surface PM accumulation has direct ecotoxicological effects. Even the study by Thimmegowda et al., 2020 can't give real proof for that as they used the correlation methods, which can't confirm causation. But I suggest reformulating your sentence. You can write "Insects accumulate PM on the body surfaces (references to the papers which confirm accumulation) and this might cause toxic effects to them (ref with confirmed effects)".

lines 47-48: first mention natural and after anthropogenic.

line 48-51: the sentence is too long and difficult to read.

lines 51-52: I am not convinced that the reference to a figure is needed in the introduction part.

line 52. differs.

line 53. Put a dot after "size" and start a new sentence.

lines 53-54: the reference to the only one case study conducted in Afghanistan is not enough to demonstrate that particle size is strongly associated with chemical composition. This is interesting of course, but either you should find more convincing references or reformulate this sentence. Also, please specify what are these PM10 and PM 2.5 (they are particles with specific diameters).

line 52: not the best choice for the reference (12 Hu et al., Characteristics and mixing state of S-rich particles...), please cite more general paper.

line 56: reference 14 (Negri et al., 2020) is actually a response to the paper by Thimmegowda et al., 2020. This is a critical response, where the authors state that the simple correlation can't reveal or confirm the toxic effects of PM on bees. This reference isn't appropriate here. Also, see my comment about the effects of PM on insects above. Rephrase this sentence as again, it is too vague and unconvincing.

line 59-60: if it isn't a particular journal style, I suggest not to have the references in the introduction part.

line 61: the sentence should be rephrased. I suggest "Currently, there is almost no information about the effects of the short-term exposure events, which happens more frequently under natural conditions".

line 64: the optimization.

lines 64-67: the sentence is too long, should be split into two.

line 70: remove "from air pollutants".

line 87: was the PM significantly more distributed on antennae in all treatment types? How do you calculate the average density on all body surfaces? How can you explain these results in terms of insect grooming behavior? Or is this related to the laboratory conditions? Did they have time to clean?

line 94: " we next asked" whom you asked? it is better to state that you were interested or you set up the question.

line 119-120: again "has a potential impact on fundamental" ... it is too speculative and unsure. Short-term exposure induces gene expression. Ok, this is possible. But you should have real confirmations for that, not only some attributions or correlations

line 124: again, which may be involved, it is too vague.

line 139: again "probably" too vague.

lines 140-142: indeed yes, it can be, but this dynamic process is dependent on many different factors.

line 150: "The impact of PM on olfactory perception is likely to be generalized across insect taxa and experienced in habitats beyond major sources of air pollution (Fig. S1.1a, b)." What do you mean by stating this? Did you imply that effects might be similar in different insects? I can argue as it might be largely dependent on the insects' grooming (self-cleaning behavior). You didn't mention or consider this factor seriously. You only mentioned that briefly in lines 156-157, but this information isn't sufficient. Although it is a critical factor to estimate the potential effects of PM pollution on insects' olfactory perception. How often do they clean themselves? Do you have this information for your study species? Again, only the assumption "although it may be less effective against particles as small as PM". But it may be effective. For PM10, it might be efficient!

line 159: diseases.

line 160: I would rather suggest using the word "cues" but not odors.

line 161: I would suggest not to cite numbers of declines and here it is the exact place where you can be more general. The papers about insect declines have been criticized and there is the opposite point of view, that the actual decline isn't that severe in numbers.

line 163: do these cited references really provide a piece of information about the effects of biodiversity decline on the ecosystem structure and functioning? To my knowledge, they just discuss rates of decline. Please, be more careful with providing references that should support your idea.

lines 163-164: reasons for insect decline are very nicely understood and well documented. It is completely wrong that they are "poorly understood". They include at the minimum climate change, chemical pollution (metals and pesticides), the spread of invasive species, habitat transformation, the spread of magnetic fields. Again, the given reference isn't the best option. You are citing the only paper indicating a potential link of pesticides to decline. Why? The picture is much more complex and they are enough sources. Just search by keywords "factors of insect decline", "factors of pollinators

decline”.

line 164: “perhaps also including the effects of air pollution odor detection”. Again, very vague.

line 167: your data doesn’t provide information about the effects of air pollution on insect pheromone synthesis.

line 169: these references are not suitable here.

line 172: (Diptera, Muscidae), what was your sample size? I saw this in the supplement but suggest briefly mentioning it here.

line 173: Air Quality Index) should be opposite Air Quality Index (AQI).

line 174: I would suggest that you move your figures from the introduction to here.

line 175: how many were males and females? Lines 212. ok, it is here, sample size should be mentioned earlier.

line 209: should this be moved to the reference list?

line 213: did you freeze-kill the insects prior to dissection, or dissected them from alive insects? How do you perform the dissection?

line 214: leg and wing... one leg and one wing per individual? Or how many?

lines 214-215: “obtain” might be not the best verb here, it has a meaning “we’ve given” better to use get, prepared, processed.

line 218: what was the thickness of your coating?

line 221: of antennae “were tiled in image J” again, not the best choice for the verb, select the other, please.

lines 223-225: “We used a more detailed scale because the commonly used PM10 and PM2.5 classifications of PM typically refer to human health and may not apply to insects.” This sentence again shows that the quality of the manuscript can be improved when authors accustom themselves to more literature sources on the topic. There are several studies on insects (I will suggest the reference list at the end of my report). All these studies use the following classification: large PM (>10 μm), coarse PM (10-2,5 μm), fine PM (2,5-0,1 μm), ultrafine PM (< 0,1 μm). If you decide to use the PM5, which isn’t widely used, provide clear references for that reason. There is information and reference about insects, and that literature confirms that the existing general classification fits insect studies too. How do you measure the diameter of your particles? How do you assess the size class in your study?

lines 225-229: how do you calculate the average density of the antennae? Did you use any software, or did you do that manually from the images? How many samples do you use for the density calculation? Did you calculate the density as a number of particles/mm²? Or was that just for the visible surface? Were that outer, central, or inner part (closer to the head) antennae? You didn’t provide any information on the resolution and scaling of your SEM images? What kind of detector do you use? For this kind of study normally the Back-Scattered Electron Detector (BSE) is used. What was the exposure period for your images? I saw the mentioned scaling in your supplements, but this is not enough.

lines 221-229: the whole passage should be redone, the estimation of PM density wasn’t performed correctly (see comments above).

line 230: what kind of linear mixed models do you use? General linear models or generalized linear mixed models? Did you check the distributions for normality of your variables of interest prior to running the statistical tests? In which program do you perform your statistical analyses? Again, this is mentioned in the supplement. But with is mentioned very briefly in the main text too.

line 232: what do you mean by “individual identity”? So, you used the id of an individual as a random factor? The classes of PM should be renamed according to general methodology

line 235: “EDX analysis was conducted by Energy Dispersive X-Ray Spectroscopy...” this should be written as “Energy Dispersive X-Ray Spectroscopy analyses (EDX) was conducted”.

line 236: What do you mean by saying this “A series of common elements previously identified in PM”? How and with which method did you previously identify a series of these elements? Or did you use the references? Indicate where did you get this previous information? Cite sources.

line 238: “on which antennae” what was your final antennae sample size?

lines 235-240: this whole passage needs more clarification, it lacks the basics of the EDX method description. “The origin of each PM was qualitatively classified according to their characteristic element composition”. This sentence is either too speculative or just wrongly formulated. I didn’t see any EDX

spectrum either in the main text, not in the supplements. They should be presented for this kind of data.

lines 271-273: this statistical part is correct.

line 280: "feet" you've meant legs?

line 281: you did that with alive insects? Did you consider the stress effect? Your EAG data might be so much affected by the stress caused by the legs and wings dissection? If it is so, then the whole results from this part are not reliable. The experiment should be repeated.

lines 323-332: what kind of primers do you use? How do you design those? For how many genes do you run qPCR? Why did you select those genes? It is very interesting to link gene expression to PM exposure, but the experiment should be designed more accurately.

lines 338-339: this is a wrong way and the method to identify the effects of PM on olfactory pathways, it is too vague and overambitious.

Now, I am not providing any comments on tables and figures because of the more general and abovementioned concerns on the methodology.

Suggested references or literature to read prior to further steps:

1. Flanders SE (1941) Dust is an inhibiting factor in the reproduction of insects. *J Econ Entomol* 34:470-472. <https://doi.org/10.1093/jee/34.3.470>
2. Grantz, D. A., Garner, J. H. B., & Johnson, D. W. (2003). Ecological effects of particulate matter. *Environment International*, 29(2-3), 213-239.
3. Kelly F, Fussell J (2012) Size, source and chemical composition as determinants of toxicity attributable to ambient particulate matter. *Atmospheric Environment* 60: 504-526
4. Łukowski A, Popek R, Jagiełło R, Mańderek E, Karolewski P (2018) Particulate matter on two *Prunus* spp. decreases survival and performance of the folivorous beetle *Geonioctena quinquepunctata*. *Environmental Science and Pollution Research* 25: 16629-16639
5. Łukowski A., Popek R., Karolewski P. (2020). Particulate matter on foliage of *Betula pendula*, *Quercus robur*, and *Tilia cordata*: deposition and ecophysiology. *Environmental Science and Pollution Research* 27: 10296-10307.
6. Osborne KH, Longcore T (2021) Effect of gypsum dust on lepidopterous larvae. *Ecotoxicology and Environmental Safety* 228: 113027
7. Papa G, Capitani G, Capri E, Pellicchia M, Negri I (2021) Vehicle-derived ultrafine particulate contaminating bees and bee products. *Science of the Total Environment* 750: 141700

Reviewer #3:

Remarks to the Author:

The study addresses the very interesting and novel question of whether particulate matter affects olfaction in an insect. The topic is highly interesting given that effects of particulate matter are well studied in humans, but very little in other organisms. The study used various approaches, ranging from scanning electron microscopy to physiological and behavioural approaches, to transcriptomics. Despite the high potential, I found that the manuscript is weak in several concerns.

1. Methodological details are missing, making it not possible to evaluate the significance and some of the (main) results.
2. In some aspects, the manuscript provides conflicting information.
3. Important statistical outcomes are not shown and the sample sizes are often unclear.
4. Some of the conclusions are not supported by the data.

All my specific comments can be found in the files attached to this review.

REVIEWER COMMENTS

Reviewer #1 (Remarks to the Author):

The authors of this study present a very interesting idea to investigate how air pollution can affect the olfactory system of insects. They combined different methods to investigate this effect from several angles. This data is novel and might have improved our understanding of air pollution effects on insect physiology and fitness. However, by using only two flight cages, the authors' experimental design cannot separate cage effects from exposure effects. The authors did not replicate the cages themselves, so the authors compared two non-independent groups of individuals to each other. They also do not mention the size of the flight cages and how many flies they put in each cage. Based on this, I think the authors cannot correctly analyse the experimental exposure data. However, the data from the wild-caught flies is still very interesting and could be written up for a more specific journal.

Response:

We thank very much for the comments. The reviewer's main concern for this study is the lack of flight cage replica, all the samples were collected from one exposure assay of only two flight cages, thus confounding the cage effect and the exposure effect. The description of our contamination procedure may have caused this misunderstanding. We completely agree with the reviewer that extra care needs to be taken for studies like this to eliminate any other possible factors that might confound the results of this experiment, including lighting, temperature, humidity, noise, or cage materials, summarised as "cage effect". Not only our flies were sampled several batches for each experiment, several other measures were taken to make sure that only the exposure effect (treatment effect) had caused olfactory functions changes: **(1)** The male and female flies were separated, randomly assigned to different flight cages each containing about 150 flies, thus our exposure was conducted in four instead of two cages of the exact same size (80cm by 40cm by 60cm) and material (cardboard boxes with five sides replaced by mosquito nets) for each assay. These cages were randomly assigned for male and female flies, then to two treatments, each flight cages were only used once to avoid any possible carry-over contaminations. We also did the experiment in two labs next to each other, with the same lighting, temperature, humidity, and noise condition, in order to keep these conditions identical between treatments. These details are now added in the methods **(Lines 225-235)**. **(2)** On the other hand, even if we conducted the exposure of each treatment in several cages, we would have encounter new problems such as position effect, in which the location of each cage will never be exactly the same in the lab. Eliminating effects like these is difficult and impractical. **(3)** Indeed, due to the complexity of our experiments, it's impossible to collect all samples in one batch (or generation). For each generation, it usually takes 3-4 days for all the adult flies to emerge, so our contamination takes places in corresponding

days (contamination batch), which is implied in our previous methods but didn't specify it clear enough. We now have specified it in **lines 252–254**. Before any experiments was started, we did several rounds of pilot tests to confirm that PM contamination on insect antennae was not coincidence, using different exposure time and particle concentration on antennal surface of houseflies and moths (see image below). For SEM imaging, we exposed the flies in different concentration of PM for several times, For EAG recordings, one batch of samples were collected for each odour type, but each experiment contains 3 contamination batches; for behavioural assays, one sample batch was taken for each odour type, each experiment also contains 3 contamination batches; for our RNA sequencing, each batch were taken in each season. Thus, not only the sampling is replicated, the male and female samples can corroborate with each other as they are from the same sample batch. (4) We appreciate that the reviewer noticed that we investigated the effects of PM from several angles. The results of our experiments, using techniques like SEM, behavioural assays, EAG recordings and RNA sequencing showed the impacts of PM from gene expression/regulation, physiological changes, behavioural response, and morphological evidences, which corroborate with each other. (5) To increase the generality of our RNA sequencing to get a more comprehensive understanding for PM impacts, we supplemented an additional contamination assay in Summer in the new version of the manuscript, which unsurprisingly corroborate with the results from Spring, and the DEGs enriched in both of the assays are validated using QPCR in the revised version of this manuscript (**Lines 131–146**).

Specific comments:

Line 173: please add how you obtained the AQIs here, i. e., move lines 208–211 up here.

Response:

We have edited as suggested.

Line 206: Did you only test this once in two cages? If yes, this means you only have one replicate for each treatment level, because flies are not independent of each other. Like this you cannot properly distinguish between the effect of

cage and the experimental treatment of pollution exposure. Or did you repeat this several times?

Response:

We appreciate this concern, the sentence in Line 206 was only referring to each time of the contamination, not for the whole study. This may have caused misunderstanding. Please see previous response. Now the sentence is changed to:

‘Both the contaminated and control flight cages were placed at locations with identical temperature and light intensity, the houseflies were supplied with sufficient water and food (except for the food odour behavioural assays, see below). This procedure was conducted in batches, and the flies were sampled within 24 h for subsequent tests (unless specified otherwise).’

Line 212: did you collect the wild flies at the same time as the exposed and control flies from your cages? Was this collection done in the vicinity of the cages? Please add this information to the method section.

Response:

Due to the highly opportunistic nature of our experiments, we rely very much on the weather conditions for sampling from the wild. Yes, the collection of wild flies in heavy pollution period were at the same period as the contaminated laboratory flies on the campus of the university, but the wild flies collected from the light and moderate pollution levels were from different pollution episodes close to the time of the exposure experiments. we cannot control for the duration of exposure of wild flies in specific weather conditions. The wild flies were collected within 200m of the cages where the exposure experiment was conducted. The information is now added, and this section is now re-written.

Line 230: please add the software used for the statistical analysis.

Response:

All the statistical analysis were conducted using JMP14. It’ s now stated in **Line 398 (section: Statistical analysis)**

Line 279-281: Does this procedure affect the EAG? Please add some reference showing that the removal of wings and feet does not affect the EAG read out. I am aware that the animal must be fixed somehow, and all flies were processed in the same manner. Nevertheless, it would be great to give the reader some idea of how invasive clipping of wings and feet is.

Response:

We appreciate this concern, but there are several lines of reasons to validate our procedure. (1) The vast majority of the insect EAG assays in literatures were conducted with the dissected insects, including some of the most important findings of insect olfactory functions published in the leading journals. Dissecting insects for EAG recording is the standard procedure in Lepidopteran, Hymenopteran and Dipteran insects. The following table contains the top most highly cited studies containing EAG recording on flies, and the vast majority of

them use dissected insects. (2) In addition, our procedure was reviewed by one of the most experienced experts on insect EAG procedures in China, Professor Yang Liu (Chinese Academy of Agricultural Sciences), who has published papers involving EAG techniques using partial insects on journals like *Molecular Biology and Evolution*, *Current Biology*, and *Molecular Ecology Resources*. (3) Indeed, using only the isolated antennae is the other common method, but the dissected insect antennae can only be viable for a small period of time. The empirical guideline in our lab is to finish all assays within 5 min before the antennae become unresponsive. Our method of keeping insects alive can significantly increase the window for the series of assays and reduce any possibility of variation due to the reduced viability. In addition, we measured the baseline of antennal response both before and after the assays, which confirms the viability of each antenna. References to the methods we use is now added. (4) We attempted the procedure using the intact insects, as suggested by the reviewer, but the wings and legs allow the flies to move around and almost impossible to obtain a steady baseline to start the assays. We are happy to repeat this assay, if the reviewer can provide some suggestions on how to conduct this procedure on houseflies.

We thank the reviewer for this suggestion of providing readers some idea of how invasive our procedure is, by comparing the recording of intact insects and the ones with wings and legs clipped, however as EAG assays should be conducted with the antennae perfectly still to obtain a relatively stable baseline, which has proved to be technically infeasible for our setup. This might be the reason why many of previous studies only used the heads, which is even more invasive than our procedure.

Title	Reference short	Journal	Year	Family	Specimen	Method
The olfactory responses of the antenna and maxillary palp of the fleshfly, Nobeliella bullata (Diptera: Sarcophagidae) and their sensitivity to blockage of nitric oxide synthase	Wasserman and Itagaki (2003)	Journal of Insect Physiology	2003	Diptera	Nobeliella bullata	whole insect
Electroantennogram, flight orientation, and oviposition responses of Aedes aegypti to the oviposition pheromone n-heneicosane	Seenivasagan et al (2009)	Parasitology Research	2009	Diptera	Aedes aegypti	head isolated
The role of volatile semiochemicals in mediating host location and selection by nuisance and disease-transmitting cattle flies	Birkett et al (2005)	Medical and Veterinary Entomology	2005	Diptera	Haematobia irritans and Musca autumnalis	head isolated
Female-biased attraction of oriental fruit fly, Bactrocera dorsalis (Hendel), to a blend of host fruit volatiles from Terminalia catappa L.	Siderhurst and Jang (2006)	Journal of Chemical Ecology	2006	Diptera	Bactrocera dorsalis	head isolated
Effect of Age on EAG Response and Attraction of Female Anastrepha suspensa (Diptera: Tephritidae) to Ammonia and Carbon Dioxide	Kendra et al (2005)	Environmental Entomology	2005	Diptera	Anastrepha suspensa	head isolated
Cucumber Volatile Blend Attractive to Female Melon Fly, Bactrocera cucurbitae (Coquillett)	Siderhurst and Jang (2010)	Journal of Chemical Ecology	2010	Diptera	Bactrocera cucurbitae	head isolated
Behavioral and electrophysiological responses of the parasitic wasp Ptyctalia concolor (Szepilgeti) (Hymenoptera: Braconidae) to Ceratitis capitata -induced fruit volatiles	Benelli et al (2013)	Biological Control	2013	Hymenoptera	Ptyctalia concolor	head isolated
Electroantennogram responses of the carrot fly, Paisa rosae , to volatile plant components	Guerin and Visser (1980)	Physiological Entomology	1980	Diptera	Paisa rosae	head isolated
Sensory and behavioural responses of the stable fly, Stomoxys calcitrans to rumen volatiles	Jeanbourequin and Guerin (2007)	Medical and Veterinary Entomology	2007	Diptera	Stomoxys calcitrans	head isolated
Electroantennogram responses of the mediterranean fruit fly, Ceratitis capitata , to a spectrum of plant volatiles	Light et al (1988)	Journal of Chemical Ecology	1988	Diptera	Ceratitis capitata	whole insect
Electroantennogram responses of the Mediterranean fruit fly, Ceratitis capitata , to the volatile constituents of nectarines	Light et al (1992)	Entomologia Experimentalis et Applicata	1992	Diptera	Ceratitis capitata	whole insect
Attraction and Electroantennogram Responses of Male Mediterranean Fruit Fly to Volatile Chemicals from Persea , Litchi and Ficus Wood	Niogret et al (2011)	Journal of Chemical Ecology	2011	Diptera	Ceratitis capitata	head isolated
Behavioral and Electroantennogram Responses of Phorid Fly Pseudacteon tricuspis (Diptera: Phoridae) to Red Imported Fire Ant, Solenopsis invicta Odor and Trail Pheromone	Chen and Fadamiro (2007)	Journal of Insect Behavior	2007	Diptera	Pseudacteon tricuspis	head isolated
Characterization of olfactory sensilla of the olive fly. Behavioral and electrophysiological responses to volatile organic compounds from the host plant and bacterial filtrate	Liscia et al (2013)	Journal of Insect Physiology	2013	Diptera	Bactrocera oleae	whole insect
Fire ant venom alkaloids act as key attractants for the parasitic phorid fly, Pseudacteon tricuspis (Diptera: Phoridae)	Chen et al (2009)	Naturwissenschaften	2009	Diptera	Pseudacteon tricuspis	head isolated
Electroantennogram responses of the stable fly, Stomoxys calcitrans , to components of host odour	Schofield et al (1995)	Physiological Entomology	1995	Diptera	Stomoxys calcitrans	whole insect
Role of plant volatiles in host plant location of the leafminer, Liriomyza sativae (Diptera: Agromyzidae)	Zhao and Kang (2002)	Physiological Entomology	2002	Diptera	Liriomyza sativae	head isolated
Chemiosensory and behavioural responses of the turnip sawfly, Athalia rosae , to glucosinolates and isothiocyanates	Barker et al (2006)	CHEMOECOLOGY	2006	Diptera	Athalia rosae	wings and legs amputated
Identification and functional analysis of a chemosensory protein from Bactrocera minax (Diptera: Tephritidae)	Cui et al (2022)	Pest Management Science	2022	Diptera	Bactrocera minax	antenna only
Identification of Volatiles From Plants Infested With Honeydew-Producing Insects and Attraction of House Flies (Diptera: Muscidae) to These Volatiles	Hung et al (2020)	Journal of Medical Entomology	2020	Diptera	Musca domestica	head isolated
Laboratory Evaluation of Natural and Synthetic Aromatic Compounds as Potential Attractants for Male Mediterranean Fruit Fly, Ceratitis capitata	Tabanca et al (2019)	Molecules	2019	Diptera	Ceratitis capitata	head isolated
Olfactory sensitivity to major, intermediate and trace components of sex pheromone in Ceratitis capitata is related to mating and circadian rhythm	Sollai et al (2018)	Journal of Insect Physiology	2018	Diptera	Ceratitis capitata	head isolated
Physiological state influences the antennal response of Anastrepha obliqua to male and host volatiles	Reyes et al (2016)	Physiological Entomology	2016	Diptera	Anastrepha obliqua	head isolated
Comparative responses of four Pseudacteon phorid fly species to host fire ant alarm pheromone and analogs	Ngumbi and Fadamiro (2015)	Chemoecology	2015	Diptera	Pseudacteon cutellatus , P. curvatus , P. obtusus	head isolated
Olfactory and behavioural responses of tabanid horseflies to octenol, phenols and aged horse urine	Baldacchino et al (2014)	Medical and Veterinary Entomology	2014	Diptera	Tabanus bromius and Aplyforus quadrispinus	antenna only

Line320: Please state here if you used 60 pairs of antennae per sample or 60 flies without antennae per sample. Or did you use both? Also, based on which parameter, the authors decided to only do three biological replicates with 180 flies/antennae pairs, rather than going for more replicates with fewer individuals per sample.

Response:

For each of the antennae transcriptome analysis, we used 60 pairs of antennae per sample and we used 60 flies without antennae per sample for body transcriptome analysis. We conducted this experiment using three replicates and 60 individuals per sample is based on both practical and analytical reasons **(1)** The results of our transcriptomic analysis have undergone a series of quality control steps to ensure the rigidity of our data. First the sequencing results were checked to make sure that the raw read counts were normalised to correct for the sequencing depths after “rubbish reads” were filtered, then the P values between each sample were calculated using hypothesis testing, which was then corrected using Benjamini/Hochberg Multiple hypothesis testing to obtain the adjusted p value. Second, the raw dataset was checked by sampling different proportions of the whole library to check for the level of representativeness. Third, and the most importantly, the Pearson’ s correlation between each sample was calculated, which confirms that the three biological replicates are consistent enough for valid results. The empirical standards are that Pearson’ s correlation over 90%, but our datasets show that they are about 99%, suggesting these samples are highly similar (see the figure below). **(2)** Three to five biological replica is a common procedure for most transcriptomic analysis involving pooled insect samples (e.g., Martin et al. 2005; Gao et al. 2020; Thimmegowda et al. 2020; Chen et al. 2021), and is the balance between the data quality and costs. Pooled samples should eliminate the individual variations and gives relatively consistent results (see attached figure 1).

Figure 1. Quality control results of transcriptome analysis, a means a very low error rate distribution along reads, b means nucleotide of G is equal to C, and A is equal to T, c means we have over 99% of high-quality reads, d means quality control results of each sample, e is the saturation curve that shows our sampling

is valid, and f is the Pearson correlation between samples.

(3) The number of individuals per replicates is based on set of pilot studies. RNA is highly unstable before they are converted into cDNA even when the whole operation is on dry ice. We need to balance the concentration of RNA and the risk of RNA degradation for the sampling. We need at least $1\ \mu\text{g}$ of total RNA to build the library. Our pilot study shows that 60 individuals should give us sufficient RNA concentration while we can keep the time of RNA sampling reasonable. We now add some of these in the Supplementary materials, and the rest are standard procedures and not usually reported in most other studies. We have added the details of sampling in methods (lines 361-371: section Transcriptome analysis and RT-qPCR).

De Vos, M., Van Oosten, V. R., Van Poecke, R. M., et al. (2005). Signal signature and transcriptome changes of Arabidopsis during pathogen and insect attack. *Molecular Plant-microbe Interactions*, 18(9), 923-937.

Gao, J., Jin, S. S., He, Y., et al. (2020). Physiological Analysis and Transcriptome Analysis of Asian Honey Bee (*Apis cerana cerana*) in Response to Sublethal Neonicotinoid Imidacloprid. *Insects*, 11(11), 753.

Thimmegowda, G. G., Mullen, S., Sottolare, K., et al. (2020). A field-based quantitative analysis of sublethal effects of air pollution on pollinators. *Proceedings of the National Academy of Sciences of the United States of America*, 117(34), 20653 - 20661.

Chen, Y. R., Tzeng, D., Ting, C., et al. (2021). Missing Nurse Bees-Early Transcriptomic Switch From Nurse Bee to Forager Induced by Sublethal Imidacloprid. *Frontiers in genetics*, 12, 665927.

References 15, 28, and 38: Reference is in all caps.

Response:

They are now changed.

Figure 1 a) please make sure you do not violate any copyright by using this map.

Response:

We did not use any online map on ArcGIS for this figure, the background map is just a shape layer, thus does not need to have a copyright claim. The data source of the PM concentration and the nature reserves are both cited.

Please name the statistical tests performed to the f-statistics given in the figure legends.

Response:

Added.

Reviewer #2 (Remarks to the Author):

General comments

The manuscript entitled “Antennal contamination from short term air pollution severely compromises insect olfactory perception” by Wang et al., represents an interesting and up-to-date effort to assess previously almost unstudied research niche. Indeed, despite the availability of the vast literature on the adverse effects of air pollution and more specifically particulate matter (PM) pollution on humans, there is less data about wildlife. The number of published research on insects is even less. This is the first reason, why the attempt of this study should be acknowledged.

Further on, the idea that a short-term air pollution episode might compromise the insects’ olfactory perception and affect the antennae sounds reasonable. The PM pollution was indeed underestimated as a risk factor for insects’ abundance and biodiversity. It wasn’ t also included in previous large-scale surveys. So, the manuscript suggests considering air pollution as a possible factor, contributing to the overall insect decline.

The multidisciplinary nature of the manuscript is also worth positive comments. The idea of using the combination of methods (SEM, EDS, EAG, qPCR, behavioural assays) is interesting and ambitious. This kind of study is needed and has the potential to improve our knowledge of insect responses to air pollution.

However, I have several concerns. First, there is a lack of understanding of the insect self-cleaning (grooming) behaviour. It is only briefly mentioned in one place of the manuscript. However, it should be taken seriously during both experimental and writing stages in this kind of research. In my experience, driven from field behavioural tests, the first body part, which insect clean immediately, is ANTENNAE. They always clean those with mouthparts or/an anterior pair of legs. Did the authors conduct any kind of self-cleaning observations for their study species under natural and laboratory conditions? My second major notice is about literature coverage. In many places, the references are either too narrow (when describing a bigger picture) or not matching. The full text of the article should be familiar to the authors to give the proper reference, not only the abstract part (see my specific comments). Indeed, the literature on the topic is scarce, but it exists! I suggested some sources at the end of my report. The other concern is methodology. I’ ve noticed several key issues in the SEM-EDX methods (use of the detector, PM density calculation). EDX spectrum is not presented in the supplementary data. It isn’ t clear why the density of PM was

higher on the antennae than on the other body parts? I can't see any reasonable explanation for that. Also, the gene-expression part sounds too vague. The primer specification wasn't clearly explained. If the EAG was performed for alive insects with dissected legs and wings, all the data obtained are compromised by huge stress factors. This is fully true, that currently, there is very limited information on the effects of PM on insects' olfactory organs. But to confirm the real effects more detailed and structured methodology is needed.

I will let the editor decide about the possibility to consider this manuscript for publication in "Nature Communications". However, all these key issues should be carefully addressed by the authors before the further step to publishing this data anywhere. This is an important study with the potential to generate high impact, but numerous issues should be corrected or redone first.

Response:

We thank very much the reviewer's comments and suggestions.

For the first concern about **grooming behaviour**, we agree that it should be discussed more in this research. Grooming may have removed some of the PM from the body of flies, but we have several lines of evidence suggesting that *it is not sufficient* to defend themselves from the impact of PM. Our procedure allows the flies to groom themselves *ad libitum* both during and after the exposure period, yet we still observed higher density of PM on the body of the contaminated flies than the clean flies. **First**, the frequency of grooming behaviour between contaminated and uncontaminated flies did not show significant differences (see Fig. S2.2), suggesting there is no significant increase in their grooming behaviour after exposing to PM. **Second**, we did not observe any significantly denser PM on their front tibia and first tarsomere than controls, suggesting that grooming may be sufficient to remove larger dirt and debris on insect bodies, but may not be sufficient to remove PM, probably due to the diameter of bristles on the tibia are larger than the size of typical PM particles (see Fig. S2.2). In fact, this was found on the legs of other species we conducted the similar exposure study, including honey bees (*Apis mellifera*), and diamond back moths (*Plutella xylostella*). **Third**, all the flies collected from the field has similar distribution of PM on their antennae to our lab flies, suggesting our exposure procedure can result in comparable level of contamination of wild flies in the field. These points are now discussed in the main text (**Lines 93-108; 182-188**) and additional results are now added in the supplementary materials. However, the key point of this study is to demonstrate PM can influence the function of the main olfactory organs of houseflies, regardless of the PM are removed or not, thus we consider extensive discussion on grooming behaviour deviates from the main scope of this study. We plan to discuss this topic more thoroughly in a focused paper on the mechanism of PM contamination on the body different insects, which we are preparing now.

We appreciate the **references** that reviewer suggested, they are now incorporated in our manuscript where appropriate (**Lines 51, 64, 66, 277**). We noticed that

except for the experiment cited by Flanders (1941), most of the existing references of similar topic focus on the impacts on insects after ingesting PM contaminated food, which is different from our current study on possibly the more common impact of direct PM exposure. These suggest that PM may influence insects via different routes and more studies are needed.

We agree completely with the reviewer that our methods need to include more details. The methods of SEM and EDX are revised as suggested (**Lines 256–288**).

Same as the reviewer, we are curious about why PM contaminate the antennae of flies than any other body parts, and conducted a follow-up study. We now understand any kind of contamination is the **combined effect** of pollutant deposition, and physical transportation (e.g. grooming). The former is the consisted of the aerodynamic interaction and the physical interaction between the PM and fly head. Our aerodynamic modelling showed that the PM can ‘peel away’ from the air flow before they encounter the antennae, and has much higher chance of deposited on the parameter of antennae than any other body parts (see attached figure 2). We also found that the antennae of flies perform differently at different scales, they are hyper-hydrophobic for large water droplets (with contact angle $> 150^\circ$) and less hydrophobic when in contact with small particles (about $5 \mu\text{m}$) using atomic force microscope (AFM). However, these results are out of the scope of this study, and will be published in a more specific paper which we are preparing now.

Figure 2. Computational Fluid Dynamic modelling results show that the PM have much higher chance of deposited on the fly antennae than any other parts of the head (manuscript in preparation).

Specific comments

line 36–37: a bit odd formulation “density increases... with”.

Response:

We are not sure what the review means for this point. Could the reviewers suggest a better expression here?

lines 43-44: Currently, the connection between 1st and 2nd sentences is weak. Probably, you can add one more sentence after the first one, briefly describing the types of anthropogenic pollutants that you mentioned at the beginning.

Response:

We have now added one sentence to highlight the impact of PM is possibly more dangerous and relatively unclear compared to other pollutants in the beginning.

lines 44-45: “may also be affected” this formulation with probability sounds too vague, if you have the proof, avoid assumptions. I know the papers you have cited here and understand why you use this “may” mode. Indeed, the literature doesn’ t really confirm that surface PM accumulation has direct ecotoxicological effects. Even the study by Thimmegowda et al., 2020 can’ t give real proof for that as they used the correlation methods, which can’ t confirm causation. But I suggest reformulating your sentence. You can write “Insects accumulate PM on the body surfaces (references to the papers which confirm accumulation) and this might cause toxic effects to them (ref with confirmed effects)”.

Response:

Changed as suggested.

lines 47-48: first mention natural and after anthropogenic.

Response:

Changed as suggested.

line 48-51: the sentence is too long and difficult to read.

Response:

This sentence is now break into two sentences.

lines 51-52: I am not convinced that the reference to a figure is needed in the introduction part.

Response:

The figure is based on our re-analysis of an existing dataset, simply to make the point that vast land surface of the world has PM concentration higher than expected. This is not well known even among the community of environment scientists who generally think PM is an urban pollutant. It was surprising even for the author of the dataset we cited, Dr. Melanie Hammer. We think it’ s a point worth presenting in Introduction, as this provides justification of why we should conduct this research. Citing a figure in the Introduction is not common but has been done in other researches, in fact, one of the papers suggested by the reviewer did it (Osborne & Longcore 2021). Can we leave this for the editors to decide whether this is appropriate for the style of this journal?

line 52. differs.

line 53. Put a dot after “size” and start a new sentence.

Response:

Changed as suggested.

lines 53-54: the reference to the only one case study conducted in Afghanistan is not enough to demonstrate that particle size is strongly associated with chemical composition. This is interesting of course, but either you should find more convincing references or reformulate this sentence. Also, please specify what are these PM10 and PM 2.5 (they are particles with specific diameters).

Response:

We have replaced this reference with a more general review papers (Kim, 2015; Yang, 2021). They both suggest that the composition of PM10 and PM2.5 are different as a general trend, despite this classification is only based on size. This result was also supported by an air sampling in Beijing, which we did not cite in the main text as this only refers to one location (Hu et al., 2016 Characteristics and mixing state of S-rich particles...) and not a global trend.

line 52: not the best choice for the reference (12 Hu et al., Characteristics and mixing state of S-rich particles...), please cite more general paper.

Response:

We have removed this reference and replaced with two review papers.

line 56: reference 14 (Negri et al., 2020) is actually a response to the paper by Thimmegowda et al., 2020. This is a critical response, where the authors state that the simple correlation can't reveal or confirm the toxic effects of PM on bees. This reference isn't appropriate here. Also, see my comment about the effects of PM on insects above. Rephrase this sentence as again, it is too vague and unconvincing.

Response:

Reference 14 is now removed and we agree with the reviewer that this is a critical response. We have rewritten this sentence to incorporate the references the reviewer suggested. We are not sure what the reviewer meant by 'this sentence as again, it is too vague and unconvincing' as we suggest that the study by Thimmegowda et al. (2020) is correlational, and only covers extremely high PM concentration (AQI over 200) over long time, such conditions are very rare in the rest of the world. Could the reviewer make any suggestions of how to improve this sentence?

line 59-60: if it isn't a particular journal style, I suggest not to have the references in the introduction part.

Response:

Again, these figures are analysed from an existing dataset, that provide important background of this study and justification of our experimental design focusing on the relatively short-term PM exposure.

The purpose of these figures is to give readers an idea of the condition and mode of PM pollution in Beijing. Thus, they are not exactly "Results" nor

should they be in Methods or Discussion. Can we leave this for the editors to decide whether this is appropriate for the style of this journal?

line 61: the sentence should be rephrased. I suggest “Currently, there is almost no information about the effects of the short-term exposure events, which happens more frequently under natural conditions”.

line 64: the optimization.

lines 64-67: the sentence is too long, should be split into two.

line 70: remove “from air pollutants”.

Response:

Changed as suggested.

line 87: was the PM significantly more distributed on antennae in all treatment types? How do you calculate the average density on all body surfaces? How can you explain these results in terms of insect grooming behaviour? Or is this related to the laboratory conditions? Did they have time to clean?

Response:

The density of PM on body parts other than antennae were calculated by taking three quadrates in each of three individuals randomly selected from the contaminated individuals. The grooming behaviour of flies is performed mostly by front tibia, which is efficient in removing dusts or debris from the insect body, and we often observe many particles on the legs of flies in our previous studies. We suspect PM may be too small or too ‘sticky’ for the existing insect cleaning mechanisms, as we did not observe increased number of PM on the front femur on the contaminated flies. All the flies should have sufficient time to clean themselves as they are allowed to groom *ad libitum* both during and after the exposure period. We totally agree with the reviewer that this topic is very important in understanding the mechanism of how PM contaminate and impact insects, but is out of the scope of this study, and should be addressed in the next paper we are preparing now.

line 94: “we next asked” whom you asked? it is better to state that you were interested or you set up the question.

Response:

We changed this sentence to “We then explored…”

line 119-120: again “has a potential impact on fundamental” … it is too speculative and unsure. Short-term exposure induces gene expression. Ok, this is possible. But you should have real confirmations for that, not only some attributions or correlations

Response:

We understand the concerns of the reviewer, however, this is the limitation of RNAseq, RT-qPCR, and pathway enrichment analysis methods. These methods, despite the fact they are almost standard procedures for studying mechanism of biological

processes now, can only demonstrate correlations at best, not concrete confirmation of the effect. Thus, these methods are commonly used to narrow down the search for important genes or pathways respond to certain treatment, which should be validated via other techniques. Our conclusion is supported by our results of KEGG pathway enrichment analysis. It is a standard knowledge base for systematic analysis of gene functions, linking genomic information with higher order functional information, this analysis will highlight the pathways that are most likely to be changed by our treatment. Now we conducted **validations** to our previous transcriptomic analyses results. **First**, we conducted additional transcriptomic analysis on samples collected in the Summer (the previous one was in the Spring). **Second**, we selected 25 overlapping DEGs from both analyses to validate their expression levels using RT-qPCR. The results shows that the DEGs in Spring and Summer corroborate with each other, and highly overlapping enriched pathways.

Indeed, the main purpose of these analyses in this paper are to demonstrate the short-term PM exposure are correlated with gene expression change in houseflies, which corroborated with the olfactory function change we observed in the previous experiments, and highlight the possibly other physiological function changes. “Real confirmations” of these effects probably requires techniques such as gene knock-outs, knock-ins, a series of functional/fitness assays that are out of the scope of this study.

line 124: again, which may be involved, it is too vague.

Response:

This part now completely re-written as suggested by another reviewer.

line 139: again “probably” too vague.

Response:

We have removed “probably” in this sentence. See the comments above.

lines 140-142: indeed yes, it can be, but this dynamic process is dependent on many different factors.

Response:

We totally agree, now this sentence is changed to “rate of which will be dependent on many factors including the frequency, duration, and concentration of air pollution episodes, the composition of PM, and the behaviour and biology of each insect species.”

line 150: “The impact of PM on olfactory perception is likely to be generalized across insect taxa and experienced in habitats beyond major sources of air pollution (Fig. S1.1a, b).” What do you mean by stating this? Did you imply that effects might be similar in different insects? I can argue as it might be largely dependent on the insects’ grooming (self-cleaning behaviour). You didn’ t mention or consider this factor seriously. You only mentioned that briefly in lines 156-157, but this

information isn't sufficient. Although it is a critical factor to estimate the potential effects of PM pollution on insects' olfactory perception. How often do they clean themselves? Do you have this information for your study species? Again, only the assumption "although it may be less effective against particles as small as PM". But it may be effective. For PM10, it might be efficient!

Response:

We have no intention to state that this effect is similar among different clades of insects. By stating "The impact of PM on olfactory perception is likely to be generalized across insect taxa and experienced in habitats beyond major sources of air pollution", we suggest that the impacts of PM pollution on insects could be larger than a single species, as the PM could impact many insect groups other than flies, and in habitats far beyond the major source of PM pollution. In the following sentences, we further explained that the PM may impact many other insect taxa, as we already detected similar PM on the body of other insect species, and may have a global impact, not only in urban area or only in developing countries like China or India. This sentence is now changed to "*PM may impact olfactory perception across different insect taxa and may be experienced in remote habitats far beyond major sources of air pollution*" to avoid confusions.

We agree with the reviewer that grooming is very important factor and needs to be addressed in detail. This part is now re-written as a separate paragraph. As we discussed previously, the process of PM contamination on the insect body is the combined effects of aerodynamic interaction, and the surface properties between the PM and the insect body as well as the grooming behaviour. These processes are too complicated to be addressed thoroughly in this paper. Please refer to our observation of grooming behaviour in our previous responses.

line 159: diseases.

line 160: I would rather suggest using the word "cues" but not odours.

line 161: I would suggest not to cite numbers of declines and here it is the exact place where you can be more general. The papers about insect declines have been criticized and there is the opposite point of view, that the actual decline isn't that severe in numbers.

Response:

Changed as suggested.

line 163: do these cited references really provide a piece of information about the effects of biodiversity decline on the ecosystem structure and functioning? To my knowledge, they just discuss rates of decline. Please, be more careful with providing references that should support your idea.

Response:

We have added more references here (Line 202).

lines 163-164: reasons for insect decline are very nicely understood and well

documented. It is completely wrong that they are “poorly understood”. They include at the minimum climate change, chemical pollution (metals and pesticides), the spread of invasive species, habitat transformation, the spread of magnetic fields. Again, the given reference isn’ t the best option. You are citing the only paper indicating a potential link of pesticides to decline. Why? The picture is much more complex and they are enough sources. Just search by keywords “factors of insect decline”, “factors of pollinators decline”.

Response:

We thank the reviewer for correcting this mistake, we now deleted any statements suggesting these effects are “poorly understood”, and have replaced some of the references.

line 164: “perhaps also including the effects of air pollution odour detection”. Again, very vague.

Response:

We were deliberately being a bit vague here as we expand the scope of our discovery by suggestions that PM may also contribute to the population decline around the world. Of course, we have no evidence at the moment but we think it’ s a direction that worth exploring. The first step is to assess how bad the PM contamination is around the world. We are now initiating a collaborative project to ask colleagues around the world to collect insect samples.

line 167: your data doesn’ t provide information about the effects of air pollution on insect pheromone synthesis.

Response:

Changed. This was an editing error.

line 169: these references are not suitable here.

Response:

We were trying to suggest another possible impact of PM pollution is by reducing population density, the population viability may be diminished, which is theoretically summarised by “Allee effect”. It describes the positive density dependence, or the positive correlation between population density and individual fitness.

line 172: (Diptera, Muscidae), what was your sample size? I saw this in the supplement but suggest briefly mentioning it here.

Response:

The sample size of each experiment was described in corresponding sections.

line 173: Air Quality Index) should be opposite Air Quality Index (AQI).

Response:

Changed as suggested.

line 174: I would suggest that you move your figures from the introduction to here.

Response:

We added the figures here, but we think the figures are more compelling presented in the Introduction.

line 175: how many were males and females?

Lines 212. ok, it is here, sample size should be mentioned earlier.

Response:

We keep them here as we think they are logically fit.

line 209: should this be moved to the reference list?

Response:

This is now moved to reference.

line 213: did you freeze-kill the insects prior to dissection, or dissected them from alive insects? How do you perform the dissection?

Response:

As mentioned previously, the samples were dried before dissection. This is specified in the previous paragraphs.

line 214: leg and wing... one leg and one wing per individual? Or how many?

Response:

We collected one set of these organs from each of five individuals. This sentence is now adapted including these details.

lines 214-215: "obtain" might be not the best verb here, it has a meaning "we've given" better to use get, prepared, processed.

Response:

Changed as suggested.

line 218: what was the thickness of your coating?

Response:

Details about our SEM procedures are now added. We were unable to measure the exact coat thickness, however it was estimated that our procedure and instrument should result in the coat thickness about 5nm, calculated by the head of the Microscopy Core Facility, Biological Technology Centre of Beijing Forestry University.

line 221: of antennae "were tiled in image J" again, not the best choice for the verb, select the other, please.

Response: This is now changed to "Antennae were stitched using Tile function in image J" (Line 271).

lines 223–225: “We used a more detailed scale because the commonly used PM10 and PM2.5 classifications of PM typically refer to human health and may not apply to insects.” This sentence again shows that the quality of the manuscript can be improved when authors accustom themselves to more literature sources on the topic. There are several studies on insects (I will suggest the reference list at the end of my report). All these studies use the following classification: large PM ($>10\ \mu\text{m}$), coarse PM ($10\text{--}2.5\ \mu\text{m}$), fine PM ($2.5\text{--}0.1\ \mu\text{m}$), ultrafine PM ($< 0.1\ \mu\text{m}$). If you decide to use the PM5, which isn't widely used, provide clear references for that reason. There is information and reference about insects, and that literature confirms that the existing general classification fits insect studies too. How do you measure the diameter of your particles? How do you assess the size class in your study?

Response:

The classification the reviewer suggested is indeed the most widely used classification of PM, however, our follow-up CFD simulated showed that PM5 interacts with insect antennae differently from PM10 and especially PM2.5 and thus should be studied separately. These results have not been published yet (see figures above). We now realise this uncommon classification may have caused confusion. This result is now re-calculated using a more conventional classification of PM, large PM ($> 10\ \mu\text{m}$), coarse PM ($10\text{--}2.5\ \mu\text{m}$), and fine PM ($< 2.5\ \mu\text{m}$). The ultrafine particle ($< 0.1\ \mu\text{m}$) is hard to quantify precisely on the antennae of flies as they are difficult to distinguish from the surface characters of antennal sensilla, or easily obscured under the tall sensillar and microtrichia, thus not included in this dataset. The revised results and methods are now in **lines 271–282**.

lines 225–229: how do you calculate the average density of the antennae? Did you use any software, or did you do that manually from the images? How many samples do you use for the density calculation? Did you calculate the density as a number of particles/mm²? Or was that just for the visible surface? Were that outer, central, or inner part (closer to the head) antennae? You didn't provide any information on the resolution and scaling of your SEM images? What kind of detector do you use? For this kind of study normally the Back-Scattered Electron Detector (BSE) is used. What was the exposure period for your images? I saw the mentioned scaling in your supplements, but this is not enough.

Response:

We did the calculation manually by counting all the visible PM on the entire antennal surface of each fly. We used a total of 120 samples in density calculation ($n = 15$ individuals). We calculated the density of PM by dividing the number of particles by the area of antennal surface, which was converted to mm^2 . The area of each antenna was calculated in ImageJ, after we stitch the whole image of antennae by ImageJ from four smaller SEM images (see Fig. 1b). We counted both antennae of each individual, one for the anterior surface, and the other for the posterior surface. We used SE detector here, and each figure was

compiled by 40 S per frame and the resolution level was 2560*1280. We have added these details in **lines 265-270**.

lines 221-229: the whole passage should be redone; the estimation of PM density wasn't performed correctly (see comments above).

Response:

This whole paragraph is now rewritten.

line 230: what kind of linear mixed models do you use? General linear models or generalized linear mixed models? Did you check the distributions for normality of your variables of interest prior to running the statistical tests? In which program do you perform your statistical analyses? Again, this is mentioned in the supplement. But with is mentioned very briefly in the main text too.

Response:

We used generalized linear mixed models here, and we have checked the distributions for normality of our variables. All the statistical analysis were conducted using JMP 14 pro. The statistical models are specified. The distribution of normality was checked before conducting statistical tests, except for the behavioural assay dictates, as they are binary distribution. These details are now stated in the Methods (**Lines 398-411**).

line 232: what do you mean by "individual identity"? So, you used the id of an individual as a random factor? The classes of PM should be renamed according to general methodology.

Response:

Yes, individuals were analysed as random factor, and the PM classes are now changed.

line 235: "EDX analysis was conducted by Energy Dispersive X-Ray Spectroscopy..." this should be written as "Energy Dispersive X-Ray Spectroscopy analyses (EDX) was conducted".

Response:

Changed as suggested.

line 236: What do you mean by saying this "A series of common elements previously identified in PM"? How and with which method did you previously identify a series of these elements? Or did you use the references? Indicate where did you get this previous information? Cite sources.

Response:

Reference is now added.

line 238: "on which antennae" what was your final antennae sample size?

Response:

The analysed particles of EDX were chosen randomly on antennal surface while we

were taking images for PM density calculation. These particles were from over 50 individuals. These are now specified in **lines 271–282**.

lines 235–240: this whole passage needs more clarification; it lacks the basics of the EDX method description. “The origin of each PM was qualitatively classified according to their characteristic element composition”. This sentence is either too speculative or just wrongly formulated. I didn’t see any EDX spectrum either in the main text, not in the supplements. They should be presented for this kind of data.

Response:

We now removed any speculations about the sources of these PM, as they are out of the scope of this study. The main purpose of this EDX analysis is to confirm that the particles we found on the body of flies are indeed airborne PM pollution, not fragmentations from their bodies or food particles. The spectrums of particles are embedded in the SEM figures in Figure S2.2.

lines 271–273: this statistical part is correct.

line 280: “feet” you’ ve meant legs?

Response:

Changed.

line 281: you did that with alive insects? Did you consider the stress effect? Your EAG data might be so much affected by the stress caused by the legs and wings dissection? If it is so, then the whole results from this part are not reliable. The experiment should be repeated.

Response:

We appreciate this concern, but there are several lines of reasons to validate our procedure. **(1)** The vast majority of the insect EAG assays in literatures were conducted with the dissected insects, including some of the most important findings of insect olfactory functions published in the leading journals. Dissecting insects for EAG recording is the standard procedure in Lepidopteran, Hymenopteran and Dipteran insects. The following table contains the top most highly cited studies containing EAG recording on flies, and the vast majority of them use dissected insects. **(2)** In addition, our procedure was reviewed by one of the most experienced experts on insect EAG procedures in China, Professor Yang Liu (Chinese Academy of Agricultural Sciences), who has published papers involving EAG techniques using partial insects on journals like *Molecular Biology and Evolution*, *Current Biology*, and *Molecular Ecology Resources*. **(3)** Indeed, using only the isolated antennae is the other common method, but the dissected insect antennae can only be viable for a small period of time. The empirical guideline in our lab is to finish all assays within 5 min before the antennae become unresponsive. Our method of keeping insects alive can significantly increase the window for the series of assays and reduce any possibility of variation due to the reduced viability. In addition, we measured the baseline of

antennal response both before and after the assays, which confirms the viability of each antenna. References to the methods we use is now added. (4) We attempted the procedure using the intact insects, as suggested by the reviewer, but the wings and legs allow the flies to move around and almost impossible to obtain a steady baseline to start the assays. This might be the reason why many of previous studies only used the heads, which is even more invasive than our procedure. We are happy to repeat this assay, if the reviewer can provide some suggestions on how to conduct this procedure on houseflies.

Title	Reference short	Journal	Year	Family	Specimen	Method
The olfactory responses of the antenna and maxillary palp of the fleshfly, Necobellera bullata (Diptera: Sarcophagidae), and their sensitivity to blockage of nitric oxide synthase	Wasserman and Itagaki (2003)	Journal of Insect Physiology	2003	Diptera	Necobellera bullata	whole insect
Electroantennogram, flight orientation, and oviposition responses of Aedes aegypti to the oviposition pheromone n-hexacosane	Seenivasagan et al (2009)	Parasitology Research	2009	Diptera	Aedes aegypti	head isolated
The role of volatile semiochemicals in mediating host location and selection by nuisance and disease-transmitting cattle flies	Birkett et al (2005)	Medical and Veterinary Entomology	2005	Diptera	Haematobia irritans and Musca autumnalis	head isolated
Female-biased attraction of oriental fruit fly, Bactrocera dorsalis (Hendel), to a blend of host fruit volatiles from Terminalia catappa L.	Siderhurst and Jang (2006)	Journal of Chemical Ecology	2006	Diptera	Bactrocera dorsalis	head isolated
Effect of Age on EAG Response and Attraction of Female Anastrepha suspensa (Diptera: Tephritidae) to Ammonia and Carbon Dioxide	Kendra et al (2005)	Environmental Entomology	2005	Diptera	Anastrepha suspensa	head isolated
Cucumber Volatile Blend Attractive to Female Melon Fly, Bactrocera cucurbitae (Coquillett)	Siderhurst and Jang (2010)	Journal of Chemical Ecology	2010	Diptera	Bactrocera cucurbitae	head isolated
Behavioral and electrophysiological responses of the parasitic wasp Plytalia concolor (Hymenoptera: Braconidae) to Ceratitis capitata -induced fruit volatiles	Benelli et al (2013)	Biological Control	2013	Hymenoptera	Plytalia concolor	head isolated
Electroantennogram responses of the carrot fly, Pila roseae , to volatile plant components	Guerin and Viser (1980)	Physiological Entomology	1980	Diptera	Pila roseae	head isolated
Sensory and behavioural responses of the stable fly Stomoxys calcitrans to natural volatiles	Jeanbourquin and Guerin (2007)	Medical and Veterinary Entomology	2007	Diptera	Stomoxys calcitrans	head isolated
Electroantennogram responses of the mediterranean fruit fly, Ceratitis capitata , to a spectrum of plant volatiles	Light et al (1986)	Journal of Chemical Ecology	1986	Diptera	Ceratitis capitata	whole insect
Electroantennogram responses of the Mediterranean fruit fly, Ceratitis capitata , to the volatile constituents of nectarines	Light et al (1992)	Entomologia Experimentalis et Applicata	1992	Diptera	Ceratitis capitata	whole insect
Attraction and Electroantennogram Responses of Male Mediterranean Fruit Fly to Volatile Chemicals from Persea , Litchi and Ficus Wood	Niogret et al (2011)	Journal of Chemical Ecology	2011	Diptera	Ceratitis capitata	head isolated
Behavioral and Electroantennogram Responses of Phorid Fly Pseudacteon tricuspis (Diptera: Phoridae) to Red Imported Fire Ant Solenopsis invicta Odor and Trail Pheromone	Chen and Fadairo (2007)	Journal of Insect Behavior	2007	Diptera	Pseudacteon tricuspis	head isolated
Characterization of olfactory sensilla of the olive fly: Behavioral and electrophysiological responses to volatile organic compounds from the host plant and bacterial filtrate	Liscia et al (2013)	Journal of Insect Physiology	2013	Diptera	Bactrocera oleae	whole insect
Fire ant venom alkaloids act as key attractants for the parasitic phorid fly, Pseudacteon tricuspis (Diptera: Phoridae)	Chen et al (2009)	Naturwissenschaften	2009	Diptera	Pseudacteon tricuspis	head isolated
Electroantennogram responses of the stable fly, Stomoxys calcitrans , to components of host odour	Schofield et al (1996)	Physiological Entomology	1996	Diptera	Stomoxys calcitrans	whole insect
Role of plant volatiles in host plant location of the leafminer, Liriomyza sativae (Diptera: Agramozidae)	Zhao and Kang (2002)	Physiological Entomology	2002	Diptera	Liriomyza sativae	head isolated
Chemosensory and behavioural responses of the turnip sawfly, Athalia rosae , to glucosinolates and isochlorogenic acids	Barker et al (2006)	CHEMOECOLOGY	2006	Diptera	Athalia rosae	wings and legs amputated
Identification and functional analysis of a chemosensory protein from Bactrocera minax (Diptera: Tephritidae)	Cui et al (2022)	Pest Management Science	2022	Diptera	Bactrocera minax	antenna only
Identification of Volatiles From Plants Infested With Honeydew-Producing Insects, and Attraction of House Flies (Diptera: Muscidae) to These Volatiles	Hung et al (2020)	Journal of Medical Entomology	2020	Diptera	Musca domestica	head isolated
Laboratory Evaluation of Natural and Synthetic Aromatic Compounds as Potential Attractants for Male Mediterranean Fruit Fly, Ceratitis capitata	Tabanca et al (2019)	Molecules	2019	Diptera	Ceratitis capitata	head isolated
Olfactory sensitivity to major, intermediate and trace components of sex pheromone in Ceratitis capitata is related to mating and circadian rhythm	Sollai et al (2018)	Journal of Insect Physiology	2018	Diptera	Ceratitis capitata	head isolated
Physiological state influences the antennal response of Anastrepha obliqua to male and host volatiles	Reyes et al (2016)	Physiological Entomology	2016	Diptera	Anastrepha obliqua	head isolated
Comparative responses of four Pseudacteon phorid fly species to host fire ant alarm pheromone and analogs	Ngumbi and Fadairo (2015)	Chemoecology	2015	Diptera	Pseudacteon cultellatus , P. curvatus , P. obtusus	head isolated
Olfactory and behavioural responses of tabanid horseflies to octenol, phenols and aged horse urine	Baldechino et al (2014)	Medical and Veterinary Entomology	2014	Diptera	Tabanus bromius and Atylotus quadrifarius	antenna only

lines 323-332: what kind of primers do you use? How do you design those? For how many genes do you run qPCR? Why did you select those genes? It is very interesting to link gene expression to PM exposure, but the experiment should be designed more accurately.

Response:

We did not run qPCR in the previous version, and now supplied these data in Figure 4 and Figure S4.2, with details about the primers now listed in Table S4.9. For qPCR, we selected 25 relatively highly expressed DEGs overlapped in both of the spring and summer antennae samples, that were involved in olfactory functions, as well as some important genes involved in self-detoxification (cytochromes) and circadian rhythm (tim). These are now specified in lines 387-397.

lines 338-339: this is a wrong way and the method to identify the effects of PM on olfactory pathways, it is too vague and overambitious.

Response:

Yes, we agree our previous way of interpreting the RNAseq dataset should be improved. Thus, we conducted an additional transcriptomic analyses sampling flies exposed to the PM in Summer, which corresponds with the previous dataset collected from Spring. We found that the expression levels of some genes and pathways overlap between the two seasons. The genes affected in both seasons, and some important genes involved in olfactory functions that enriched in one season are selected for qPCR. This paragraph is now re-written to incorporate the additional results.

Now, I am not providing any comments on tables and figures because of the more general and abovementioned concerns on the methodology.

Suggested references or literature to read prior to further steps:

- 1. Flanders SE (1941) Dust is an inhibiting factor in the reproduction of insects. J Econ Entomol 34:470 - 472. <https://doi.org/10.1093/jee/34.3.470>*
- 2. Grantz, D. A., Garner, J. H. B., & Johnson, D. W. (2003). Ecological effects of particulate matter. Environment International, 29(2-3), 213-239.*
- 3. Kelly F, Fussell J (2012) Size, source and chemical composition as determinants of toxicity attributable to ambient particulate matter. Atmospheric Environment 60: 504 - 526*
- 4. Łukowski A, Popek R, Jagiełło R, Mąderek E, Karolewski P (2018) Particulate matter on two Prunus spp. decreases survival and performance of the folivorous beetle Geonioctena quinquepunctata. Environmental Science and Pollution Research 25: 16629-16639*
- 5. Łukowski A., Popek R., Karolewski P. (2020). Particulate matter on foliage of Betula pendula, Quercus robur, and Tilia cordata: deposition and ecophysiology. Environmental Science and Pollution Research 27: 10296-10307.*
- 6. Osborne KH, Longcore T (2021) Effect of gypsum dust on lepidopterous larvae. Ecotoxicology and Environmental Safety 228: 113027*
- 7. Papa G, Capitani G, Capri E, Pellicchia M, Negri I (2021) Vehicle-derived ultrafine particulate contaminating bees and bee products. Science of the Total Environment 750: 141700*

Response:

We thank the reviewer for these important references, they are now incorporated into the manuscript in Introduction and Discussion parts.

Reviewer #3 (Remarks to the Author):

The study addresses the very interesting and novel question of whether particulate matter affects olfaction in an insect. The topic is highly interesting given that effects of particulate matter are well studied in humans, but very little in other organisms. The study used various approaches, ranging from scanning electron microscopy to physiological and behavioural approaches, to transcriptomics. Despite the high potential, I found that the manuscript is weak in several concerns.

- 1. Methodological details are missing, making it not possible to evaluate the significance and some of the (main) results.*
- 2. In some aspects, the manuscript provides conflicting information.*
- 3. Important statistical outcomes are not shown and the sample sizes are often unclear.*
- 4. Some of the conclusions are not supported by the data.*

All my specific comments can be found in the files attached to this review.

Response:

We thank the reviewer very much to provide such detailed comments, these are truly very important to improve the manuscript. Details are now added the details in the Methods as suggested, explained and clarified where seem to caused confusion, added statistical outcomes and sample size as required, and corrected the conclusions according to the statistics. For the convenience of further editing, our responses to each of the comments are attached with the original files, but the modified version of the manuscript is loaded separately.

Reviewers' Comments:

Reviewer #1:

Remarks to the Author:

The authors have substantially improved their manuscript and precisely answered my comments. However, it is still unclear how many batches, i.e. pairs of cages, were used in the experiment. If this was not adequately replicated, the authors cannot call this an experiment. Also, it is not clear whether batch was used as a random effect in their stats analysis.

Reviewer #2:

Remarks to the Author:

The manuscript entitled "Short term air pollution contamination from severely compromises insect antennal olfactory perception" has been now significantly improved. The novelty of the paper is not subject to dispute. Indeed, particulate matter (PM) is very rarely studied among insects, and the effects on the olfactory system haven't been investigated before. I fully agree with the authors' statement that PM might be an additional, however yet underestimated factor, of insects decline. So far, systematic reviews for the factors of this global issue do not refer to case studies probably because there are not that many of those. To summarize, this research is novel and is in high demand by scientific society. I appreciate the great effort made by authors since the first submitted version and am thankful for carefully addressing all critical comments! In my opinion, after addressing several issues, this work can be accepted for publication!

Comments to the Introduction part

In the title, you should correct "short term" to "short-term"

Lines 48-49: I have a question about the formulation of the first sentence and the selection of references here. "The detrimental impacts of anthropocentric pollutants to organism health and fitness, and thus population viability have been extensively documented for vertebrate wildlife (references 1-4)". The cited references describe the problems with pollution in: 1) marine microalgal forest (wildlife but not vertebrate), 2) in Accipitriformes birds, 3) in a very "broad sense" (from sediments to human) and about detrimental effects of plastic (of course, it is important, but a bit confusing), 4) in birds again. I see a problem with the logic here and suggest reformulating a sentence and correcting references. Could it be formulated like this:

Suggestion: "The detrimental impacts of anthropocentric pollutants on organism health, fitness, and population viability have been extensively documented for wildlife – from plants to vertebrates".

With such formulation, the choice for the references could be logically justified.

Lines 49-51: We can't say yet that PM has more severe effects than gaseous pollutants, as data isn't enough (and that's why your study is important!). Again, I suggest reformulating the sentence a bit. Also, "the jump" to "insects' topic" in the next sentence is a bit too sharp now.

Suggestion: "Particulate matter (PM) might be even more dangerous than other common air pollutants such as NO_x or ozone, however, its ecotoxicological effects remain unclear for many types of organisms including insects".

Lines 51-52: To your information, there is a recently published a paper in STOTEN about PM accumulation in wasps "Mobile samplers of particulate matter – Flying omnivorous insects in detection of industrial contamination", it might be relevant to your literature review.

Lines 53-54: Remove "e.g." from the parenthesis. It is better to be concise.

Line 60: Should the reference be at the end of the sentence?

Lines 59-62: It might be not that straightforward, I would add "There is evidence that PM10 has contains more inorganic or metal components including toxic heavy metal elements, and PM2.5 contains more organic pollutants such as benzene and polycyclic aromatic hydrocarbons".

Line 63: Correct to "The effects of particles on insect the reproduction function of insects were first documented in the..."

Line 68: Start a new sentence after "functions". Shorter sentences are better!

Lines 70-75: This is an important note!

Comments to the Results part

Line 103: In our study with wasps, we also found that in wild populations PM was significantly more present around antennae openings than in the other body parts.

Lines 107-108: I fully agree with this finding.

Line 109: Then, we (remember words order)

Lines 132: why Spring and Summer are capitalized?

Comments to the Discussion part

Line 165: Remove the word "combined" just start "Our experiments reveal..."

Line 167: Start new sentence after "ordours". "This finding highlights a potentially severe impact of short-term, sub-lethal PM exposure to insects".

Line 175: Our EAG analysis confirms (add -s ending)

Line 189: Correct to "Grooming behaviour (self-cleaning) including the antennae surface purifying is widespread in insects (ref. 38). But, in flies, it might not protect antennae from PM contamination"

Lines 196-197: I fully agree with this comment!

Lines 200-202: There is a review paper about the effects of heavy metals on different groups of economically important terrestrial insects, that paper might spark some ideas "Ecotoxicological effects of heavy metal pollution on economically important terrestrial insects".

Note about air pollution and chemical perception. These references, might be useful Direct and indirect effects of chemical contaminants on the behaviour, ecology and evolution of wildlife | Proceedings of the Royal Society B: Biological Sciences (royalsocietypublishing.org)

Or e.g. this one

Ozone pollution disrupts plant-pollinator systems - ScienceDirect

Comments to the Materials and methods part

Line 271: What kind of SEM detector was used?

Line 277: was that mm² or μm² ?

Thank you for your work and good luck in science!

Reviewer #3:

Remarks to the Author:

The revised version of the manuscript is much better than the original submission. Thanks to the authors for improving the manuscript. Still, there are several issues to be considered: e.g. the title is not meaningful, as is true for several sentences of the text, and some data need to be presented differently to being conclusive. All my comments can be found in the attached documents.

Reviewer #1 (Remarks to the Author):

The authors have substantially improved their manuscript and precisely answered my comments. However, it is still unclear how many batches, i.e. pairs of cages, were used in the experiment. If this was not adequately replicated, the authors cannot call this an experiment. Also, it is not clear whether batch was used as a random effect in their stats analysis.

We regret not describing our experiments more clearly, but treatment (polluted vs control) is not confounded with either batches (pairs of cages) or laboratory spaces. We used new containers to collect the treatment (ambient Beijing air) or control (filtered air) flies and discarded the used containers. Covid restrictions meant that we had to conduct our treatment of flies in several locations in Beijing, and we randomised the allocation of laboratory space to treatment or control at these different locations. The morphological, EAG, behavioural experiments and transcriptomic analyses include individuals from a total of 35 batches, with multiple batches for each experiment. We have edited the methods to make this clearer [**lines 254-255**] and provide a new table [**Supplementary materials Table S5.1**] (also see the original datasets:

https://osf.io/v92xe/?view_only=0744a389866a4cdcb602ae967fe61960). Finally, we had already indicated that we included 'batch' as a random effect in the previous version [**Lines 414-420, Table S3.2, S3.4 and subsequently in the supplementary materials**], and there was no significant batch effect in these analyses.

Reviewer #2 (Remarks to the Author):

The manuscript entitled "Short term air pollution contamination from severely compromises insect antennal olfactory perception" has been now significantly improved. The novelty of the paper is not subject to dispute. Indeed, particulate matter (PM) is very rarely studied among insects, and the effects on the olfactory system haven't been investigated before. I fully agree with the authors' statement that PM might be an additional, however yet underestimated factor, of insects decline. So far, systematic reviews for the factors of this global issue do not refer to case studies probably because there are not that many of those. To summarize, this research is novel and is in high demand by scientific society. I appreciate the great effort made by authors since the first submitted version and am thankful for carefully addressing all critical comments! In my opinion, after addressing several issues, this work can be accepted for publication!

Thank you

Comments to the Introduction part

In the title, you should correct "short term" to "short-term"

Corrected [**lines 1**]

Lines 48-49: I have a question about the formulation of the first sentence and the selection of references here. “The detrimental impacts of anthropocentric pollutants to organism health and fitness, and thus population viability have been extensively documented for vertebrate wildlife (references 1-4)”. The cited references describe the problems with pollution in: 1) marine microalgal forest (wildlife but not vertebrate), 2) in Accipitriformes birds, 3) in a very “broad sense” (from sediments to human) and about detrimental effects of plastic (of course, it is important, but a bit confusing), 4) in birds again. I see a problem with the logic here and suggest reformulating a sentence and correcting references. Could it be formulated like this:

Suggestion: “The detrimental impacts of anthropocentric pollutants on organism health, fitness, and population viability have been extensively documented for wildlife – from plants to vertebrates”.

With such formulation, the choice for the references could be logically justified.

Thank you, we have changed the sentence accordingly [lines 48-49]

Lines 49-51: We can't say yet that PM has more severe effects than gaseous pollutants, as data isn't enough (and that's why your study is important!). Again, I suggest reformulating the sentence a bit. Also, “the jump” to “insects' topic” in the next sentence is a bit too sharp now.

Suggestion: “Particulate matter (PM) might be even more dangerous than other common air pollutants such as NO_x or ozone, however, its ecotoxicological effects remain unclear for many types of organisms including insects”.

Thank you, we have changed the sentence accordingly [lines 49-52]

Lines 51-52: To your information, there is a recently published a paper in STOTEN about PM accumulation in wasps "Mobile samplers of particulate matter – Flying omnivorous insects in detection of industrial contamination", it might be relevant to your literature review.

Paper now included [lines 52]

Lines 53-54: Remove “e.g.” from the parenthesis. It is better to be concise.

Changed as suggested [lines 54-55]

Line 60: Should the reference be at the end of the sentence?

Changed as suggested [line 61]

Lines 59-62: It might be not that straightforward, I would add “There is evidence that PM₁₀ has contains more inorganic or metal components including toxic heavy metal elements, and PM_{2.5} contains more organic pollutants such as benzene and polycyclic aromatic hydrocarbons”.

Changed as suggested [line 61]

Line 63: Correct to “The effects of particles on insect the reproduction function of insects were first documented in the...”

Changed as suggested [lines 64-65]

Line 68: Start a new sentence after “functions”. Shorter sentences are better!

Changed as suggested [lines 69]

Lines 70-75: This is an important note!

Comments to the Results part

Line 103: In our study with wasps, we also found that in wild populations PM was significantly more present around antennae openings than in the other body parts.

Thank you for the information, we have similar findings.

Lines 107-108: I fully agree with this finding.

Line 109: Then, we (remember words order)

Changed as suggested [lines 111]

Lines 132: why Spring and Summer are capitalized?

No longer capitalised [lines 134] and all following

Comments to the Discussion part

Line 165: Remove the word “combined” just start “Our experiments reveal...

Changed as suggested [lines 167]

Line 167: Start new sentence after “ordours”. “This finding highlights a potentially severe impact of short-term, sub-lethal PM exposure to insects”.

Changed as suggested [lines 169]

Line 175: Our EAG analysis confirms (add -s ending)

Changed as suggested [lines 177]

Line 189: Correct to “Grooming behaviour (self-cleaning) including the antennae surface purifying is widespread in insects (ref. 38). But, in flies, it might not protect antennae from PM contamination”

Largely changed as suggested, but removed ‘purifying’ [lines 191-192]

Lines 196-197: I fully agree with this comment!

Lines 200-202: There is a review paper about the effects of heavy metals on different groups of economically important terrestrial insects, that paper might spark some ideas “Ecotoxicological effects of heavy metal pollution on economically important terrestrial insects”.

Note about air pollution and chemical perception. These references, might be useful Direct and indirect effects of chemical contaminants on the behaviour, ecology and evolution of wildlife | Proceedings of the Royal Society B: Biological Sciences (royalsocietypublishing.org)

Or e.g. this one

Ozone pollution disrupts plant–pollinator systems – ScienceDirect

We are aware of the review paper, but in the interests of limited space, we would rather not discuss the effects of heavy metals, which are different to the effects described in this manuscript

Comments to the Materials and methods part

Line 271: What kind of SEM detector was used?

We have consulted with the Hitachi engineers, the secondary electron detector (SE) for Hitachi does not have a model number. The catalogue number of this SE is 539-0315, which is not usually reported for this type of equipment.

Line 277: was that mm² or μm² ?

It is mm² [lines 287], also see [line 284]

Thank you for your work and good luck in science!

Reviewer #3 (Remarks to the Author):

The revised version of the manuscript is much better than the original submission. Thanks to the authors for improving the manuscript. Still, there are several issues to be considered: e.g. the title is not meaningful, as is true for several sentences of the text, and some data need to be presented differently to be conclusive. All my comments can be found in the attached documents.

Supplementary information

Page 3: The position of at least c and d differed, as only in d are Sensilla placodea visible

The image (d) includes the edge of the ventral side. For clarity we now use different images, so that all are of the ventral side.

We now replaced these figures. [Fig. S1.2]

Page 9: What should this be.

The c2 should be χ^2 . Now corrected. [Table. S3.2]

Main text

Title: the word from has been removed, and the title now reads: "Short term particulate matter contamination severely compromises insect antennal olfactory perception"

Line 89. We have removed "wear and tear" [line 83]

Line 94: We'd rather retain this important sentence, which we have edited, viz: "... may not be possible for insects, such as bees and flies, that do not have antennal scales" [line 88]

Line 124: We have changed it to respond. [line 144]

Line 128. In the interests of space, we prefer to leave these multiple statistics in the figure legend.

Line 148-159: we'd rather leave this concluding sentence here, since it summarises a component of our research. However, we are happy to be guided by the editor.

Line 152-154: we'd rather leave this introductory sentence here, since it provides the justification of a new component of our research

Line 154-: now reads: "In the summer, PM impacts the antennae of males more severely than females: 898 ..." **[lines 137-138]**

Line 172: now edited: "... only one differentially expressed gustatory/odorant receptor gene (GR2) was found in both male and female antennae" **[lines 152-153]**

Line 181: now changed to "many of these CYPs were downregulated in either male or female fly antennae..." **[line 162]**

Line 187: our combined experiments (y-maze assays and EAGs) shows that it's more likely that the flies are unable to detect the odours, but our results are unable to conclude whether or not the flies are unable to respond to odours, which involves the processing of higher-order neurones, so we prefer to leave the sentence as it is. **[line 169]**

Line 209: Now edited: "potentially widespread impact of air pollution on insect pheromone and cue perception, which will affect mating and foraging success. For example, even sub-lethal ..." **[lines 209-210]**

Line 218: we'd rather retain the more cautious "may be". **[line 200]**

Line 220: Manufacturer of air purifier now identified **[line 222]**

Line 220: SEM now defined (Scanning Electron Microscope) **[lines 223-224]**

Line 223-225: We have edited the sentence to make the meaning clearer: "We removed the deodorisation component of the air filter, thereby ensuring, as far as possible, that gaseous pollutants (other than PM) in the laboratory environment were similar to that outside the building." **[lines 237-238]**

Line 244: Now edited: "Contaminated flies were exposed to the outside ambient air in a laboratory with open windows in various locations in Beijing ..." **[lines 248-249]**

Line 266: We have images from the lab control group, on which almost no PM was found, and from the wild flies, which shows the similar trend as the lab-contaminated, but we only presented the results of the lab-contaminated ones on Fig.1. This sentence is now edited: "We additionally obtained the head, thorax, abdomen, a front leg, and a wing of each of five individuals from the contaminated laboratory group." **[lines 272-275]**

Line 276: Now edited: "visible PM on the antennal surface, defined by the image, and divided by the area of each antenna ..." **[lines 282-284]**

Line 278: It's OK, since the glass fibre filter is about 5 cm in diameter, and PM are deposited on it randomly and evenly.

Line 286: We have consulted with the Hitachi engineers; the Energy Dispersive X-Ray Spectroscopy for Hitachi SEM attachment does not have a model number and is generally not reported.

Line 291: We are happy to remove this referral to the Figure, but we thought it was helpful to the reader.

Line 336: This sentence is changed to 'The odour stimulations were delivered at the air speed of 0.5 m/s and duration of 0.5 s.' **[Lines 344-345]**

Line 338: The position of the steel nozzle is kept constant throughout the experiments at 2 cm from the antennae. We moved the nozzle up a bit for picture purpose only, otherwise the antennae would be almost invisible from the picture. Yes, the steel nozzle has a hole and the Pasteur pipette was inserted each time before generation odour impose. See attached picture. We now added a reference to the protocol we used.

Line 361: This sentence is changed as suggested and the formular updated. **[Line 363]**

Line 380: It's standard procedure. This sentence is now removed. **[Line 378]**

Line 412: sentence has been edited: "...to explain, for each odour and sex, the variation in the relative blank value" **[Lines 417-418]**

Line 582: The label detection zone is removed. **[Fig. 2a]**

Line 611: We have changed the values to positive.

Line 622: It's now specified. **[Line 625]**

Line 639: Fig. 4 shows the how PM influence the gene expression of housefly ANTENNAE, and Fig. S4.2 shows the gene expression of housefly BODIES.

Reviewers' Comments:

Reviewer #1:

Remarks to the Author:

The authors have now addressed all my comments and I think the manuscript is now fit for publication

Reviewer #3:

Remarks to the Author:

The manuscript is mostly in excellent shape now, but there are still some issues that are not scientifically sound and need to be corrected:

Legend Fig S1.2: please delete the sentence "All the images were taken of the ventral side of the flagellum of the antennae, where most of the sensilla were found." This is still not true: different sides of the honeybees and wasp antennae were used. Also, I do not think that flies have most of the sensilla on the ventral side of their antenna.

Line 248: "Contaminated flies were exposed for 12 h to the outside ambient air in a laboratory with open windows in various locations in Beijing (Table S5.1), on days with high AQI levels (101-150), ambient temperatures at 15-25°C, and between 7 am to 7 pm when the flies are most active." This sentence is still unclear; how can a single laboratory be in various locations?

254: please use "...with similar size,..."

348: please remove the reference to Fig. S2.2

418: "to explain the variation in the relative blank value": this part of the sentence is still not meaningful. Food and mating odors are analysed, I guess, and not "relative blank values".

Fig. 2 b-d: as stated before, it makes no sense to plot the yellow dashed lines as the minimum proportion required for a significant deviance from 50% depends on the number of flies that did a choice. According to the authors, 60 flies were tested each, however, some flies did not respond and were discarded. Thus, the number of flies that made a decision was <60, and it would be very surprising if the number was always the same. Assuming that the number of flies that did a decision varied among dilution levels, the minimum proportion required for a significant deviance from 50% would differ among dilution levels. This is important to resolve.

Tables S2.4 and S2.5: Please replace " $P > F$ " by " P ". Only p-values are shown in the column

Table S2.6: please specify in the title, what the 4 different p values indicate; further, only one of the first two p-values is needed (p or p.adj); please also check the spelling in the first line of the column

Table S3.2: Please replace " $P > X^2$ " by " P ". Only p-values are shown in the column

Table S3.4: Please replace " $P > t$ " by " P ". Only p-values are shown in the column

Reviewer #1 (Remarks to the Author):

The authors have now addressed all my comments and I think the manuscript is now fit for publication

Response: we thank the reviewer's valuable suggestions and efforts which significantly improved the quality of this manuscript.

Reviewer #3 (Remarks to the Author):

Legend Fig S1.2: please delete the sentence "All the images were taken of the ventral side of the flagellum of the antennae, where most of the sensilla were found." This is still not true: different sides of the honeybees and wasp antennae were used. Also, I do not think that flies have most of the sensilla on the ventral side of their antenna.

Response: now deleted.

Line 248: "Contaminated flies were exposed for 12 h to the outside ambient air in a laboratory with open windows in various locations in Beijing (Table S5.1), on days with high AQI levels (101-150), ambient temperatures at 15-25°C, and between 7 am to 7 pm when the flies are most active."

This sentence is still unclear; how can a single laboratory be in various locations?

Response: this sentence has changed to: the outside ambient air in treatment laboratories with open windows in various locations in Beijing...

254: please use "...with similar size,..."

Response: changed.

348: please remove the reference to Fig. S2.2

Response: we did not reference to Fig. S2.2 here, presumably there is an error?

418: "to explain the variation in the relative blank value": this part of the sentence is still not meaningful. Food and mating odors are analysed, I guess, and not "relative blank values".

Response: now this is changed to: Tukey post hoc test to explain the variation in relation to blank value

Fig. 2 b-d: as stated before, it makes no sense to plot the yellow dashed lines as the minimum proportion required for a significant deviance from 50% depends on the number of flies that did a choice. According the authors, 60 flies were tested each, however, some flies did not respond and were discarded. Thus, the flies that made a decision was <60, and it would be very surprising if the number was always the same. Assuming that the number of flies that did a decision varied among dilution levels, the minimum proportion required for a significant deviance from 50% would differ among dilution levels. This is important to resolve.

Response: we thank the reviewer for raising these concerns, we agree that the declines

are misleading, and are now removed. However, we think there is a misunderstanding in the procedure used in these assays. For the behavioral assays, we discard the unresponsive individuals while kept doing the assays until the designed number of replications for each batch is reached, thus the number of replications is the same for each assay. This is stated in our Methods, and in our original dataset that is now available online.

Tables S2.4 and S2.5: Please replace “P>F” by “P”. Only p-values are shown in the column
Response: now removed (Table S3 and S4).

Table S2.6: please specify in the title, what the 4 different p values indicate; further, only of the first two p-values is needed (p or p.adj); please also check the spelling in the first line of the column

Response: we agree, only the first two p values are needed, thus we only kept these columns. The meaning of the p.adj is specified in the title.

Table S3.2: Please replace “P>X2” by “P”. Only p-values are shown in the column

Response: now removed (Table S6).

Table S3.4: Please replace “P>t” by “P”. Only p-values are shown in the column

Response: now removed (Table S7).